# Comparative analysis of two paradigm bacteriophytochromes reveals opposite functionalities in two-component signaling

Elina Multamäki [1], Rahul Nanekar[2], Dmitry Morozov [3], Topias Lievonen[2], David Golonka [4], Weixiao Yuan Wahlgren[5], Brigitte Stucki-Buchli [2], Jari Rossi [1], Vesa P. Hytönen [6,7], Sebastian Westenhoff[5], Janne A. Ihalainen [2 ✉], Andreas Möglich [4] & Heikki Takala [1,2 ✉]

Bacterial phytochrome photoreceptors usually belong to two-component signaling systems which transmit environmental stimuli to a response regulator through a histidine kinase domain. Phytochromes switch between red light-absorbing and far-red light-absorbing states. Despite exhibiting extensive structural responses during this transition, the model bacteriophytochrome from *Deinococcus radiodurans* (DrBphP) lacks detectable kinase activity. Here, we resolve this long-standing conundrum by comparatively analyzing the interactions and output activities of DrBphP and a bacteriophytochrome from *Agrobacterium fabrum* (Agp1). Whereas Agp1 acts as a conventional histidine kinase, we identify DrBphP as a light-sensitive phosphatase. While Agp1 binds its cognate response regulator only transiently, DrBphP does so strongly, which is rationalized at the structural level. Our data pinpoint two key residues affecting the balance between kinase and phosphatase activities, which immediately bears on photoreception and two-component signaling. The opposing output activities in two highly similar bacteriophytochromes suggest the use of light-controllable histidine kinases and phosphatases for optogenetics.

[1] Faculty of Medicine, Anatomy, University of Helsinki, Helsinki, Finland. [2] Department of Biological and Environmental Science, Nanoscience Center, University of Jyvaskyla, Jyvaskyla, Finland. [3] Department of Chemistry, Nanoscience Center, University of Jyvaskyla, Jyvaskyla, Finland. [4] Lehrstuhl für Biochemie, Universität Bayreuth, Bayreuth, Germany. [5] Department of Chemistry and Molecular Biology, University of Gothenburg, Gothenburg, Sweden. [6] Faculty of Medicine and Health Technology, BioMediTech, Tampere University, Tampere, Finland. [7] Fimlab Laboratories, Tampere, Finland. ✉email: janne.ihalainen@jyu.fi; heikki.p.takala@jyu.fi

Two-component signaling systems are mainly found in prokaryotes and allow cells to respond to environmental signals[1]. These systems have been under extensive research ever since their discovery, as they control a wide range of cellular mechanisms from enzymatic activity to transcription regulation[2]. A canonical two-component system consists of a homodimeric sensor histidine kinase (HK) and its cognate response regulator (RR)[3]. To the extent it has been studied, most HK proteins sense chemical signals and generally reside within the plasma membrane[4]. The output activity is exerted by an intracellular HK module, consisting of two subdomains: a dimerization histidine phosphotransfer (DHp) domain, and a catalytic ATP-binding (CA) domain. Based on their DHp sequence and according to Pfam, the HK proteins can be divided into five subtypes, called HisKA, HisKA_2, HWE_HK, HisKA_3, and His_kinase[4,5]. The subjects of the current study, the bacteriophytochromes from *Deinococcus radiodurans* (DrBphP) and *Agrobacterium fabrum* (Agp1), both fall within the HisKA family.

The HK catalyzes autophosphorylation and subsequent phosphotransfer to the cognate RR. During the autophosphorylation reaction, the eponymous histidine of the DHp domain is phosphorylated[6,7], either within the same monomer (*cis*) or the sister molecule of the homodimer (*trans*)[8]. In the phosphotransfer reaction, the phosphate is relayed to a conserved aspartate residue within a receiver (REC) domain of the RR. This reaction entails RR activation and elicits output responses such as altered gene expression[3,6,9].

HKs may also act as phosphatases that hydrolyze the phosphoaspartyl bond in the phosphorylated response regulator, thus resetting the two-component system[10,11]. Whereas the kinase activity has been extensively studied[12], the importance of the phosphatase activity has been appreciated more recently[13–15]. In two-component systems, a dynamic balance between kinase and phosphatase activities determines the net output and downstream physiological effects. The underlying kinase-active and phosphatase-active conformational states are necessary for balancing the output activity of the two-component system[16,17].

In contrast to the typical transmembrane HK receptors, light-sensitive receptors are frequently soluble. This facilitates their structural and mechanistic analyses[4,18,19]. As a case in point, phytochromes are red/far-red light-sensing photoreceptors that regulate diverse physiological processes in plants, fungi, and bacteria, e.g., chromatic adaptation and phototaxis in prokaryotes[20,21]. Plant phytochromes exert downstream physiological responses via light-dependent interactions with partner proteins, nucleocytoplasmic shuttling and protein degradation[22,23]. By contrast, bacterial phytochromes (BphPs) usually belong to two-component signaling systems, with a cognate response regulator commonly encoded in the same operon[18,24,25]. BphPs contain an N-terminal photosensory module (PSM), divided into PAS (period/ARNT/single-minded), GAF (cGMP phosphodiesterase/adenylyl cyclase/FhlA) and PHY (phytochrome-specific) domains[20]. The PSM binds a biliverdin IXα chromophore via a thioether linkage to its conserved cysteine within the PAS domain[26,27]. The PSM is followed by a C-terminal output module, most commonly a HK domain.

Photoactivation by red and far-red light drives biliverdin *Z/E* isomerization, which underlies the phytochrome switch between its red light-absorbing (Pr) and far-red light-absorbing (Pfr) states[24]. In darkness, phytochromes can thermally revert to their resting state which is the Pr state in canonical phytochromes[28]. As first demonstrated for the model bacteriophytochrome DrBphP from *D. radiodurans*, light induces extensive structural changes in the photosensory module that are relayed to the output module[29].

In the cyanobacterial phytochrome Cph1[30–33] and several bacteriophytochromes[34,35], the dark-adapted Pr state exhibited

higher kinase activity than the Pfr state. In particular, the Agp1 bacteriophytochrome from *A. fabrum* (also known as AtBphP1, based on the former species designation *A. tumefaciens*) displays histidine kinase activity in its resting Pr state[28,36]; in the Pfr state, the autophosphorylation and phosphotransfer reactions are downregulated by 2-fold and 10-fold, respectively[28]. The kinase activity of Agp1 has been shown to control bacterial conjugation[37]. Although DrBphP has been implicated in the control of carotene production[21], no kinase activity has been demonstrated for DrBphP, notwithstanding close sequence homology and the elaborate structural changes this receptor undergoes under light[24,29]. Despite the eminent role of DrBphP as a paradigm for photoreception, the enzymatic activity and hence the exact physiological role of this model phytochrome have hence remained enigmatic.

Here, we unravel this long-standing puzzle by studying the enzymatic activity and interactions of DrBphP and Agp1, as two canonical bacteriophytochromes with HK effector domains. By pursuing an integrated biochemical and structural strategy, we show that despite close homology, Agp1 acts as a histidine kinase whereas DrBphP functions as a light-activated phosphatase. Our biochemical and structural data pinpoint two key residues proximal to the catalytic histidine that affect the balance between the kinase and phosphatase activities. Together, the two phytochromes provide soluble, light-controllable systems with opposite activities for the study and application of two-component signaling.

## Results

**The dark reversion of DrBphP is affected by DrRR**. We employed UV-vis absorption spectroscopy to investigate whether the cognate response regulators interact with DrBphP and Agp1 and potentially affect the photoactive states of these bacteriophytochromes. For reference, we also generated a hybrid receptor, denoted as Chimera, which comprises the DrBphP PSM and the Agp1 HK domain (Fig. 1a). DrBphP, Agp1, and Chimera all showed typical absorption spectra with Soret and Q-band absorption peaks for both the Pr and Pfr states, which were unaffected by the addition of the cognate RR (Fig. 1b). The thermal reversion of phytochrome samples after applying saturating red light (655 nm) exhibited multiple exponential phases in all cases, irrespective of the presence of the RR (Supplementary Fig. 1a). The recovery of Agp1 was faster than that of DrBphP, while that of the Chimera was between those of DrBphP and Agp1. Earlier studies indicated that the dark reversion in phytochromes is affected by the dimerization interfaces in both the PSM and HK domain[38]. In line with this notion, the spectral characteristics of the Chimera are evidently governed by both the Agp1 HK and DrBphP PSM.

The dark reversion kinetics of Agp1 and Chimera were unaffected by the response regulator from *Agrobacterium fabrum* (AtRR1), but that of DrBphP was significantly accelerated by *Deinococcus radiodurans* response regulator (DrRR). This finding indicates that DrRR binds to the DrBphP HK, thereby favoring the Pr state conformation. Interestingly, this contrasts with the *Arabidopsis thaliana* phytochrome B, where the binding of the phytochrome-interacting factor (PIF) stabilizes the Pfr state[39].

**DrBphP interacts with DrRR more strongly than Agp1 does with AtRR1**. To further analyze the interaction between the phytochromes and their RRs, we applied size-exclusion chromatography on fluorescently labeled RR proteins (Fig. 2a). Indicative of binding, the addition of DrBphP caused a shift in the retention of EGFP-labeled DrRR towards lower volumes. The observed interaction was independent of the photoactivation of

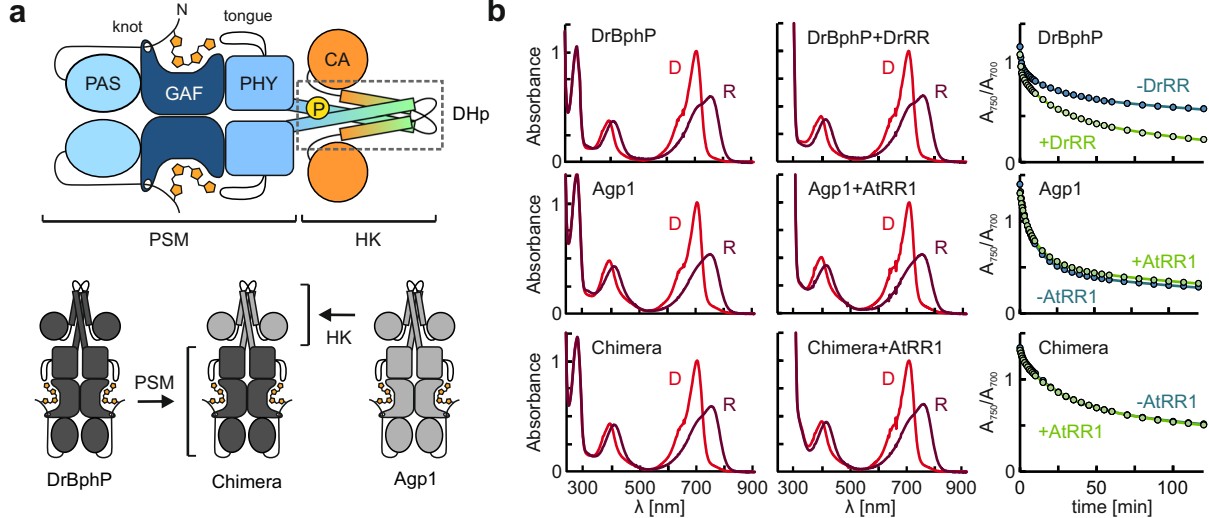

**Fig. 1 Overall architecture and UV-vis spectroscopy of DrBphP, Agp1, and their Chimera with and without their cognate response regulator. a** Schematic representation of a canonical bacteriophytochrome with a histidine kinase (HK) effector domain. The site of the phosphorylated histidine is indicated as the letter P. In addition, a schematic presentation of the phytochrome chimera is shown, where the photosensory module (PSM) of DrBphP is combined with the HK domain of Agp1. Abbreviations: Period/ARNT/single-minded (PAS), cGMP phosphodiesterase/adenylyl cyclase/FhlA (GAF), phytochrome-specific (PHY), histidine kinase (HK), dimerization Histidine phosphotransfer domain (DHp), catalytic ATP-binding domain (CA). **b** The absorption spectra of the BphP HKs with and without their cognate response regulators (RR) in dark (D) or under red light (R). The right-most panels show their dark reversion kinetics as an $A_{750}/A_{700}$ ratio over time, where 0 min corresponds to the time the 655-nm illumination ceased. Source data are provided as a Source data file.

DrBphP. By contrast, the EGFP-AtRR1 retention was not significantly affected by the presence of Agp1, suggesting no or a weak interaction between Agp1 and AtRR1 (Fig. 2a). See Supplementary Fig. 1c, d for additional measurements.

To further investigate the BphP/RR interactions, we resorted to surface plasmon resonance (SPR). The changes in the SPR signal were measured for the response regulator immobilized on the SPR chips while flowing the phytochromes across the sensor surface. The binding of DrRR to DrBphP in the Pr state was evaluated from the steady-state saturation signal (Fig. 2c), resulting in a dissociation constant $K_D$ of $(43 \pm 8)\,\mu M$ when using a 1:1 molar binding model (Supplementary Fig. 2i). This value was verified by Langmuir kinetic analysis which yielded an affinity of comparable strength ($K_D \sim 10\,\mu M$). A lower signal amplitude and a $K_D$ value of $(60 \pm 7)\,\mu M$ for the DrBphP/DrRR pair were observed upon red-light application. Notably, the slightly weaker binding in the Pfr state concurs with the above spectroscopic measurements where DrRR binding favors the Pr state. Consistent with the SEC analysis, the interaction between Agp1 and AtRR1 was substantially weaker, and the binding curve did not reach saturation at the highest achievable Agp1 concentration of 154 $\mu M$. We hence estimated the affinity to be on the order of hundreds of micromolar. The shape of the SPR response graph indicates that the association and dissociation kinetics of the Agp1/AtRR1 are fast, which precluded the kinetic evaluation. That notwithstanding, the Agp1/AtRR1 interaction was not notably affected by red light (Supplementary Fig. 1b).

As a complementary method, we applied isothermal calorimetry (ITC). The DrBphP/DrRR interaction could be described by a 1:1 molar binding model with a $K_D$ of $(8.1 \pm 1.3)\,\mu M$ (Fig. 2b). Unlike in a blue light-regulated HK[40], this interaction was not affected by the addition of the ATP analog AMP-PNP (Supplementary Fig. 2b). Agp1 binding to AtRR1 could not be reliably detected by ITC (Fig. 2b), consistent with the SEC data and the fleeting binding seen in SPR. The binding parameters were similar in a different buffer condition (Supplementary Fig. 2a, i). Furthermore, cross-interaction was neither detected

between DrBphP and AtRR1 nor between Agp1 and DrRR (Supplementary Fig. 2g).

Taken together, the interactions of DrBphP and Agp1 with their cognate response regulators were clearly different. Next we studied whether these differences correlate with enzymatic activity, as we speculated that the function of these phytochromes is reflected in their interactions.

**Agp1 functions as a histidine kinase but DrBphP acts as a phosphatase.** We characterized the kinase activity of the bacteriophytochromes by $^{32}P$-$\gamma$-ATP autoradiography (Fig. 3a). The autophosphorylation reaction of Agp1 occurred preferably in the Pr state and was reduced under red light illumination, consistent with previous reports[36]. If AtRR1 was present, it received a phosphate from Agp1 in the phosphotransfer reaction. This reaction occurred preferably in the dark-adapted Pr state but was almost absent under constant red-light illumination (i.e., in the Pfr state). This verifies that Agp1 binds to and transfers its phosphate to AtRR1 in its kinase-active Pr state.

Intriguingly, DrBphP lacked autokinase or phosphotransfer activities in both the Pr and Pfr states (Fig. 3a). The absence of kinase activity is surprising as the DrBphP and its PSM evidently undergo light-induced structural changes that seem to be conserved among other phytochromes[29,41,42]. Moreover, all homologous bacteriophytochrome HKs studied to date exhibited light-dependent kinase activity. The unusual absence of kinase activity in DrBphP could in principle be due to (1) lack of interaction with the DrRR; (2) inability of its PSM to transduce signals to the HK effector; or (3) inactivity of the DrBphP HK module. Scenario 1 can be ruled out according to the above results, which consistently showed interaction between DrBphP and DrRR. To address scenario 2, we assessed the histidine kinase activity of the Chimera and found it to function similarly to the wild-type Agp1 with robust autokinase and phosphotransfer activity in the Pr state, but reduced activity in the Pfr state (Fig. 3a). This result states that DrBphP undergoes productive

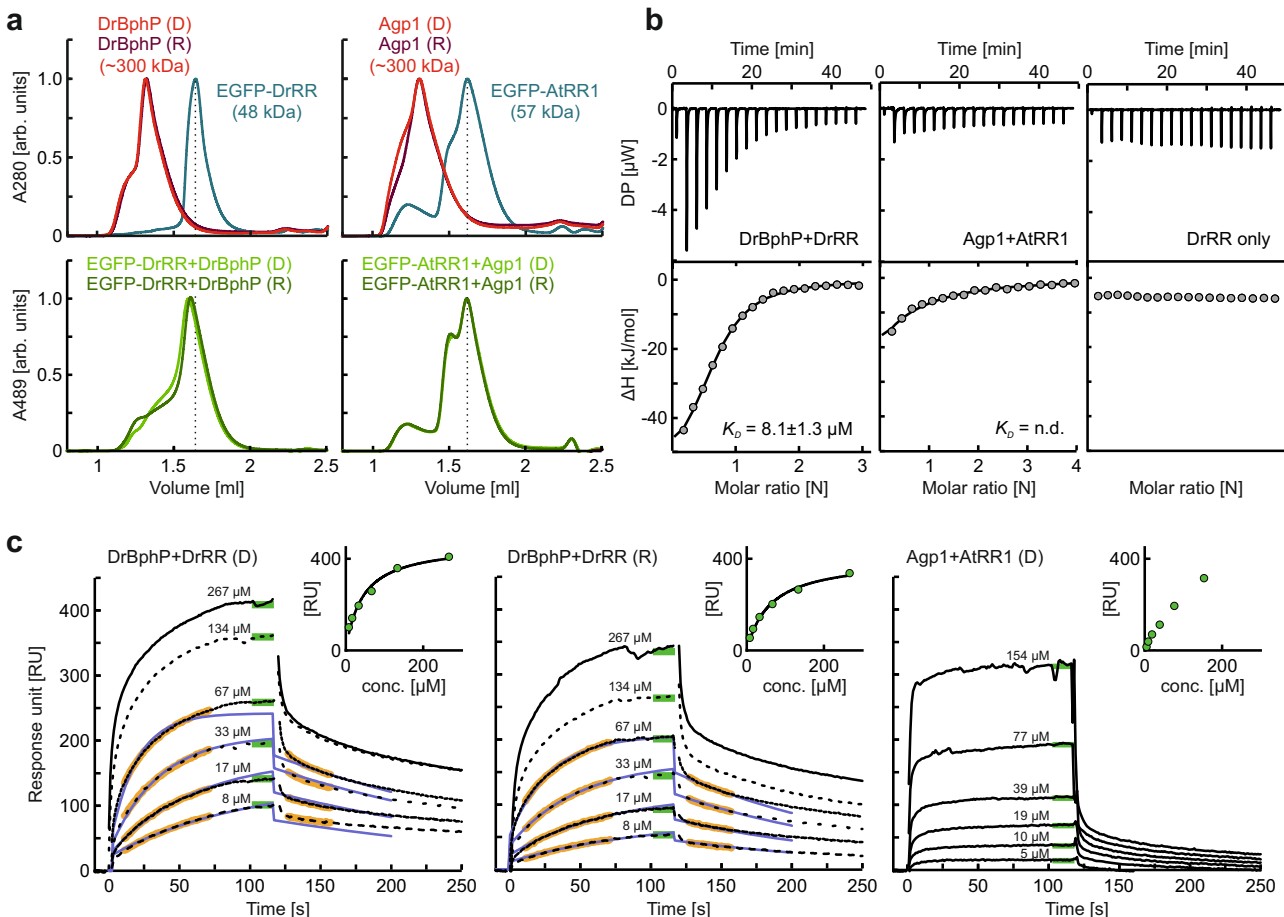

**Fig. 2 Quantitative analyses of the BphP/RR interactions. a** Size-exclusion chromatography (SEC) of the EGFP-labeled response regulators DrRR and AtRR1 in the absence (top) and presence (bottom) of DrBphP and Agp1, respectively. Vertical dashed lines indicate the retention volume of free RR monomer. Top panels: In isolation, EGFP-DrRR (45.4 kDa) and EGFP-AtRR1 (45.4 kDa) eluted as a monomer and a monomer/dimer mixture, respectively. DrBphP (84.0 kDa) and Agp1 (83.8 kDa) are known to be dimers, and their apparently high molecular weights (~300 kDa) can be explained by the poor resolution of large proteins in the conditions. Bottom panels: When combined with DrBphP in its dark-adapted state (D) or after 655-nm illumination (R), the profile for EGFP-DrRR shifted to shorter retention times, indicative of interactions. By contrast, addition of Agp1 had little effect on the retention of AtRR1. The top panels are plotted at 280 nm and the bottom panels at 489 nm. The A489 signals from BphPs were negligible. **b** Isothermal titration calorimetry (ITC) measurements. Differential power (DP) resulting from injections of the response regulator to the BphP is plotted against time, and the binding enthalpy (ΔH) is plotted against the molar ratio of the proteins. See Supplementary Fig. 2 for additional data and control measurements. **c** Surface plasmon resonance (SPR) measurements. The response regulators were coupled on the sensor surface, and varying concentrations of the corresponding phytochrome were applied in darkness (D) or after red-light illumination (R). The sensorgrams (black lines), the kinetic fits (blue lines), and the parts of the data that were used for kinetic analysis (orange) are indicated. For kinetic analyses, DrBphP concentrations of 8–67 µM were used. Green lines mark the $R_{eq}$ values that were used for evaluating the steady-state affinity data, shown in Supplementary Fig. 2i. Inset: Steady-state fit of the concentration series, where $R_{eq}$ values were used for affinity approximation. See Supplementary Fig. 2i for a table of SPR fitting values. Source data are provided as a Source data file.

structural changes that are conducive to controlling HK activity, thereby ruling out scenario 2.

To address scenario 3, Phos-tag gels were applied where unphosphorylated proteins and their phosphorylated counterparts are resolved based on migration through the gel matrix. In this analysis, unphosphorylated and phosphorylated response regulators were clearly separated from another (Fig. 3b). The assay confirmed that the wild-type Agp1 phosphorylates AtRR1 preferably in the Pr state and revealed that it cross-phosphorylates DrRR with similar efficiency (Fig. 3b). However, like in the radiolabeling assay (Fig. 3a), DrBphP lacked kinase activity, as it could not produce phosphorylated DrRR (phospho-DrRR).

The residue immediately following the catalytic histidine, denoted as H + 1, is acidic in the majority of sensor histidine kinases and has been implicated in the autophosphorylation

reaction[12]. Whereas Agp1 has Asp529 in this position and thus conforms to the prevalent sequence motif, DrBphP unusually possesses a histidine in the corresponding position 533 (Fig. 3g). To test the role of the H + 1 position, we generated the Agp1 D529H and DrBphP H533D variants. The D529H mutation rendered Agp1 inactive (Fig. 3b), thus verifying the importance of this acidic residue for the kinase activity. Like the wild-type DrBphP, the H533D variant appeared inactive (Fig. 3b, Supplementary Fig. 4c). Therefore, this single mutation in the H + 1 position is insufficient to rescue the kinase activity of DrBphP.

As sensor histidine kinases may also function as phosphatases[10,43,44], we tested the DrBphP and Agp1 HKs in that regard. Of particular advantage, the Phos-tag gels allow to assess the dephosphorylation of phospho-RR proteins. To this end, we generated the phosphorylated response regulators chemically by treatment with acetyl phosphate[45]. DrRR was

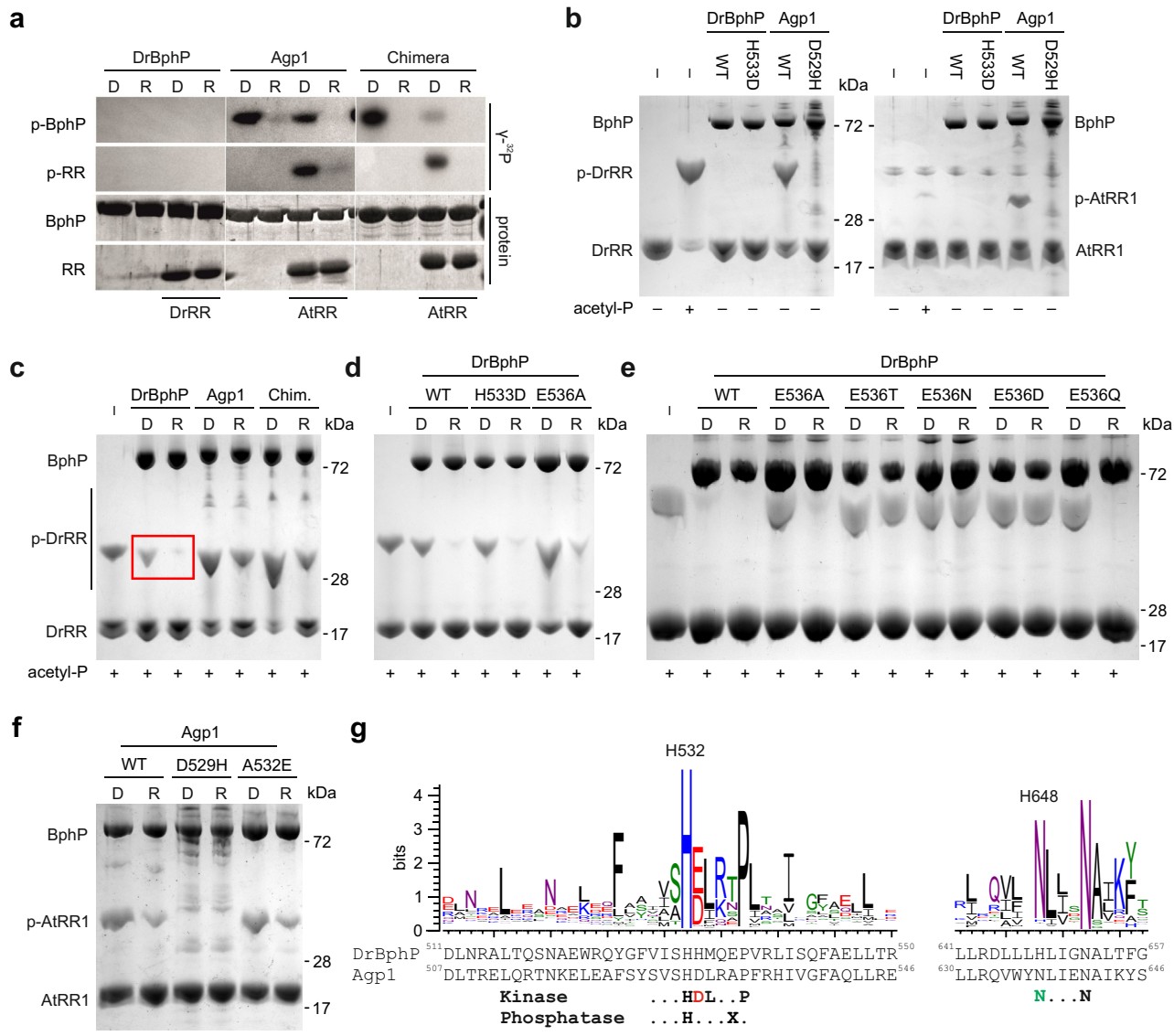

**Fig. 3 Kinase and phosphatase activity of DrBphP and Agp1. a** Kinase and phosphotransfer activity of the phytochromes (BphP), detected for radioactive phosphate (γ-$^{32}$P) and total protein. Each phytochrome sample was incubated with γ-$^{32}$P-ATP, either with or without the response regulator (RR), DrRR in case of DrBphP, and AtRR1 in case of Agp1 and Chimera. Extended gels with molecular weight marker positions are shown in Supplementary Fig. 3. **b** Kinase activity in darkness of DrBphP, Agp1, and their variants with the H+1 residue mutated. Each well was loaded with equal amounts of response regulator, all reactions contain ATP, and the total protein amount is visualized by protein staining. The phosphorylated response regulators (denoted p-DrRR and p-AtRR1) migrate more slowly in the gels and are therefore resolved from their unphosphorylated counterparts. **c** Phos-tag detection of the phosphatase activity of DrBphP (red box), Agp1, and Chimera. Equal amounts of phospho-DrRR were applied to each reaction. The letters D and R denote reactions performed in darkness or under red light, respectively. See Supplementary Fig. 4a for an extended gel. **d** Phosphatase activity of DrBphP and its variants H533D and E536A. Equal amounts of phospho-DrRR were applied to each reaction. See Supplementary Fig. 4b for an extended gel. **e** Phosphatase activity of additional DrBphP exchanges at the H+4 position. **f** Kinase activity of Agp1 and its variants D529H and A532E. See also Supplementary Fig. 4h. **g**. Sequence logo of 250,000 histidine kinase sequences, shown here for the H box around the phospho-accepting histidine (H532 in DrBphP) and the N box in the CA subdomain. The height of each letter indicates the amount of conservation of the corresponding amino acid (one-letter code). The protein sequences of DrBphP and Agp1, and the fingerprint sequence motifs are shown below the graph. An aspartate residue in the H+1 position is deemed important for histidine kinase activity[12]. In the case of phosphatase activity, the determinant residue at the H+4 position[14] varies and is denoted as X. D = dark sample; R = red-illuminated sample. Positions of molecular weight markers are shown in panels (**b**–**f**), and all measurements of panels (**a**–**f**) have been repeated independently at least three times. Source data are provided as a Source data file, which include full versions of the gels.

phosphorylated robustly, whereas AtRR1 responded to the treatment weakly. Phospho-DrRR was then incubated together with ATP and either DrBphP or Agp1. In the reactions, net phosphatase activity would decrease the amount of phospho-

DrRR, whereas net kinase activity would increase it. As expected, addition of Agp1 or Chimera led to an increase in phospho-DrRR when incubated in darkness, indicating kinase activity of these proteins (Fig. 3c). By contrast, the addition of DrBphP decreased

the amount of phospho-DrRR, especially upon red-light exposure (red box in Fig. 3c). These findings reveal that DrBphP acts as a phosphatase with higher activity in the Pfr state than in the Pr state. The DrBphP apoprotein was unresponsive to light and appeared similar in activity to Pr-state DrBphP (Supplementary Fig. 4f). The H533D mutation did not alter the phosphatase activity (Fig. 3d), and DrBphP was incapable of de-phosphorylating phospho-AtRR1 (Supplementary Fig. 4i).

Interestingly, the phosphatase activity in DrBphP depended on ATP addition as phospho-DrRR levels remained unchanged in the absence of ATP (Supplementary Fig. 4a), which is consistent with other studies[46]. In the presence of ADP, the DrBphP phosphatase activity was greatly decreased, and it was altogether absent if ATP was replaced with GTP (Supplementary Fig. 4f).

The residue in the H + 4 position is important for phosphatase activity of HisKA family proteins[14]. Indeed, the corresponding residue E536 in DrBphP appeared to have a role in the reaction as its mutation to alanine reduced the phosphatase activity (Fig. 3d). The E536A variant maintained somewhat higher phospho-DrRR amounts, not only in the Pfr but also in the Pr state, potentially by shielding the phospho-DrRR from spontaneous hydrolysis during the reaction. Exchanges of the same residue to threonine, asparagine, and aspartic acid (E356T, E536N, and E536D) abolished the phosphatase activity completely (Fig. 3e), thus underlining the importance of the H + 4 position for the phosphatase activity. By contrast, when changing Glu536 to the structurally similar glutamine (E536Q), the phosphatase activity was preserved. Like the wild-type DrBphP, none of the H + 4 variants showed any histidine kinase activity (Supplementary Fig. 4c, e). Likewise, the opposite exchange in Agp1 of the H + 4 alanine to glutamic acid (A532E) did not affect the net kinase/phosphatase activity compared to the wild-type HK (Fig. 3f).

**DrRR crystal structure reveals a canonical response regulator dimer.** As DrBphP and Agp1 strikingly differed in their enzymatic activity and interactions, we next asked whether these differences could be explained by the structure of the interface between the DHp and RR. To model this interface with confidence, we solved the crystal structure of the response regulator from *D. radiodurans* (DrRR) at 2.1 Å resolution (see Table 1 for crystallographic statistics). The protein, which consists only of a receiver (REC) domain, crystallized in the tetragonal P4₁2₁2 space group with four monomers in the asymmetric unit. These monomers form two inverted 4-5-5 dimers with a dimerization interface built by the α4–β5–α5 face of each monomer[47] (Fig. 4a), similar to most other phytochrome RR structures with a REC domain[48–50]. However, the homologous AtRR1 assumes an arm-in-arm REC dimer[35], in which the C-terminal extension forms an antiparallel β-strand interface with a sister monomer (Fig. 4a). Overall, the structure of the DrRR is highly similar to other reported response regulators. It contains the structural features and the conserved residues critical for its receiver function in two-component signaling (Fig. 4c). These structural details along with functional results (Fig. 3) verify that DrRR can function as a canonical response regulator in a two-component signaling system.

Notably, the crystal structure of DrRR contained $Ca^{2+}$ instead of $Mg^{2+}$ ions found in other response regulator structures[35,48–50]. The $Ca^{2+}$ ions played a central role in this crystal form, as their replacement with $Mg^{2+}$ did not allow crystal formation. $Ca^{2+}$ ions occupied the active site of the DrRR in a similar way to $Mg^{2+}$ in the AtRR1 structure[35]. Although $Ca^{2+}$ is chemically similar to $Mg^{2+}$, its larger size leads to diffuse coordination of the ion in the active sites (Fig. 4b) and 45% higher B-factors compared to $Mg^{2+}$ ions modeled at the same sites. Consequently, the $Ca^{2+}$

interactions differ between the four monomers in the asymmetric unit, being most similar to AtRR1 in monomer A[35]. In each case, the $Ca^{2+}$ ions are hexagonally coordinated to surrounding atoms, which involve water molecules, the side chains of Glu15, Asp16, Asn17, the phospho-accepting Asp66, and the main-chain oxygen of Asn68 (Fig. 4b).

Given its presence in the DrRR crystal structure, we tested the effects of $Ca^{2+}$ in the DrBphP activity and DrRR binding. We discovered that although the DrBphP/DrRR interaction was slightly stronger in the presence of $Ca^{2+}$ (Supplementary Fig. 2h), the DrBphP enzymatic activity was lost (Supplementary Fig. 4f, g).

**Complex models show different interactions in DrBphP and Agp1.** To analyze how the interplay of the HK and RR proteins impacts on two-component signaling, we prepared models for the DrBphP/DrRR and Agp1/AtRR1 pairs. Given the lack of high-resolution structural data on phytochrome HK domains, we generated homology models based on the complex structure of *Thermotoga maritima* HK853 (3DGE)[9] and the crystal structures of the DrRR and AtRR1[35] (Fig. 5). To assess the physiological relevance of the structural models and the binding interfaces, we performed a covariance analysis of cognate HK/RR pairs[51–53]. Prior covariance analyses assigned cognate HK/RR pairs based on genomic proximity. By contrast, we focused on a set of hybrid receptors which comprise HK and RR moieties in a single polypeptide chain, thus allowing to assign interacting, cognate HK/RR pairs with high confidence. The multiple sequence alignment of several thousand such receptors revealed strong residue covariation not only within the HK and RR parts individually but also in between them[54,55]. As in the previous

| Table 1 Crystal data collection and processing statistics. | |
|---|---|
| *Data collection* | |
| Space group | P 4₁ 2₁ 2 |
| Cell dimensions | |
| *a, b, c* (Å) | 87.65, 87.65, 181.21 |
| *α, β, γ* (°) | 90.00, 90.00, 90.00 |
| Resolution (Å) | 49.74-2.0 (2.15-2.10)ᵃ |
| $R_{merge}$ | 0.172 (2.882) |
| $CC_{1/2}$ | 0.999 (0.503) |
| $I/\sigma(I)$ | 12.01 (1.09) |
| Completeness (%) | 100.0 (100.0) |
| Redundancy | 4.13 (3.94) |
| Wilson B factor | 49.54 |
| *Refinement* | |
| Resolution (Å) | 49.74-2.10 (2.15-2.10)ᵃ |
| No. of reflections | 39,970 (2891)ᵃ |
| $R_{work}/R_{free}$ | 0.181/0.218ᵇ (0.330/0.318) |
| Overall B factor | 59.84 |
| No. of atoms | |
| Protein | 4,389 |
| Heterogenᶜ | 9 |
| Water | 240 |
| *Geometry* | |
| RMSD | |
| Bond lengths (Å) | 0.013 |
| Bond angles (°) | 1.811 |
| Ramachandran | |
| Favored (%) | 96 |
| Allowed (%) | 16 |
| Outliers (%) | 5 |
| PDB Code | 6XVU |

ᵃOuter shell values used in the refinement are in parentheses.
ᵇTest set for $R_{free}$ calculation constitutes 5% of total reflections that were randomly chosen.
ᶜThis includes nine $Ca^{2+}$ atoms.

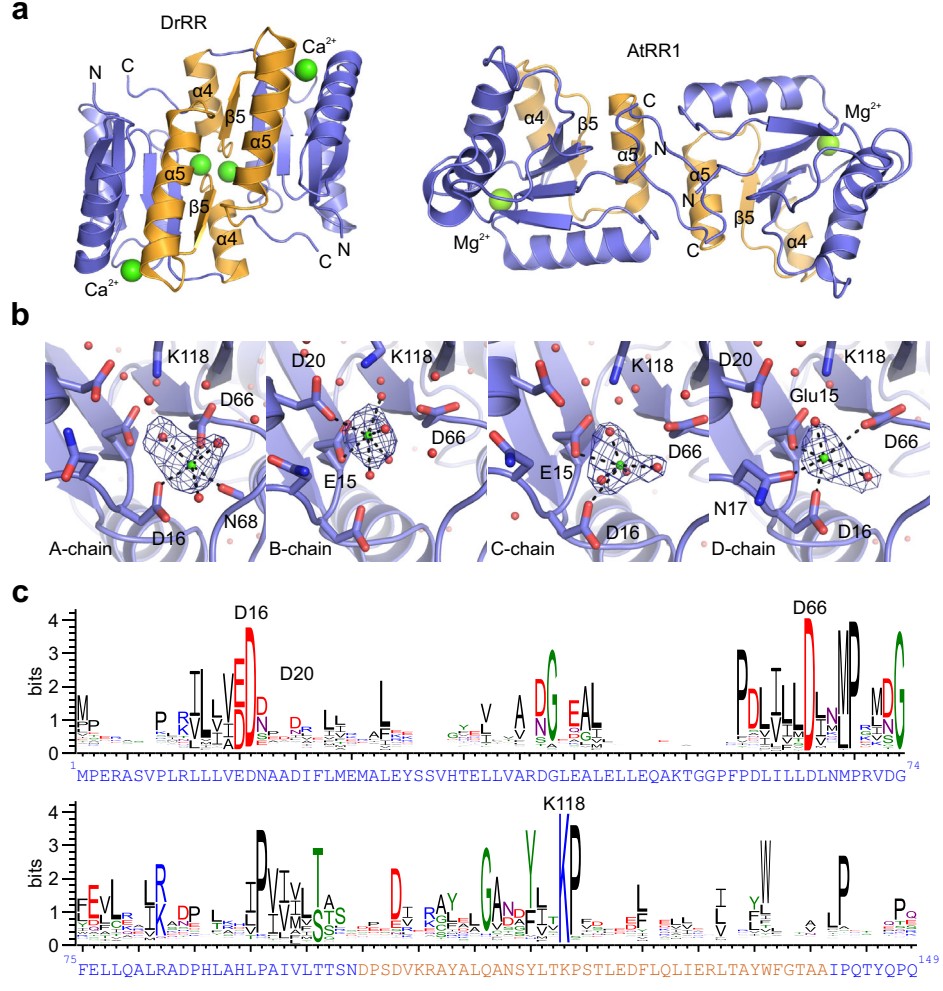

**Fig. 4 Crystal structure of the *Deinococcus radiodurans* response regulator (DrRR). a** Cartoon representation of the dimeric DrRR and AtRR1 structures (PDB code 5BRJ for AtRR1[35]). Both response regulators only consist of receiver (REC) domains. The α4–β5–α5 face of each response regulator monomer is shown in orange, and the rest of the protein in blue. Ca$^{2+}$ and Mg$^{2+}$ ions at the active sites, the N- and C-termini, as well as the dimerization helices are marked. In the case of DrRR, the dimer formed by chains A and B is shown. **b** The active site of DrRR with its Ca$^{2+}$ ions and interacting residues. The localization of the Ca$^{2+}$ ions (green) and their interactions (black dashed lines) differ between the chains. The omit difference (F$_o$–F$_c$) map of the Ca$^{2+}$ ions is shown as blue mesh at 5.0σ. The omit maps were calculated for each monomer by repeating the final refinement step without the Ca$^{2+}$ ion and the coordinating water molecules. **c** Sequence logo derived from 50,000 response regulator sequences. The height of each letter indicates the amount of conservation for the corresponding amino acid (one-letter code). The key DrRR residues are shown above the graph, and the full amino acid sequence of DrRR is given below the graph with the same coloring as in panel (**a**).

analyses, significant inter-domain covariance was observed for pairs of certain residues in the DHp domain and the RR (Supplementary Fig. 7a, b). When mapped on the presently generated structural model of the complex, strong pairwise residue covariation likewise localized to the HK/RR interface (Supplementary Fig. 7), speaking for realistic complex models.

To address the stability of the complex models in solution, we conducted classical molecular dynamics (MD) simulations at 300 K, 1 atm. pressure, and 0.1 M NaCl using the Gromacs molecular dynamics package[56]. Over a 200 ns trajectory, both the DrBphP/DrRR and Agp1/AtRR1 complexes were stable. The RMSD equilibration times for the protein backbone atoms were around ~60 ns for the DrBphP/DrRR complex and ~80 ns for Agp1/AtRR1 (Supplementary Fig. 5), suggesting that the interactions are more defined and stronger in the DrBphP/DrRR complex. Starting from the 100 ns time point of the trajectory, we extracted snapshots at 10 ns intervals and analyzed their residue interactions. Representative snapshots are shown in Fig. 5, all snapshots are given in Supplementary Fig. 5a, b.

Overall, the interactions between Agp1 and AtRR1 were transient and more variable than the ones in the *D. radiodurans* pair, as gauged by the larger overall RMSD values between successive time steps of the simulation and by higher mobility of the protein backbone atoms throughout the MD trajectory (Supplementary Fig. 5c). Analysis of the snapshots in the PISA server[57] revealed that both complex interfaces have hydrophobic core regions. The average solvation free energy upon formation of the interface indicated this interface to be more extensive in the DrBphP/DrRR complex (−47.3 kJ/mol) than in the Agp1/AtRR1 complex (−24.3 kJ/mol).

The simulations suggest that the RRs interact mainly through their α1 helix (aa. 18–32 in DrRR) that aligns with the helical bundle of the four DHp helices. In addition to this main interface, the DrRR showed interactions via a loop region (aa. 119–121) that connects strand β5 and helix α5. Notably, the position of this β5–α5 loop and the length of the α5 helix differed between DrRR and AtRR1, thus allowing DrRR more extended interactions with its phytochrome partner. The complexes contain polar interactions

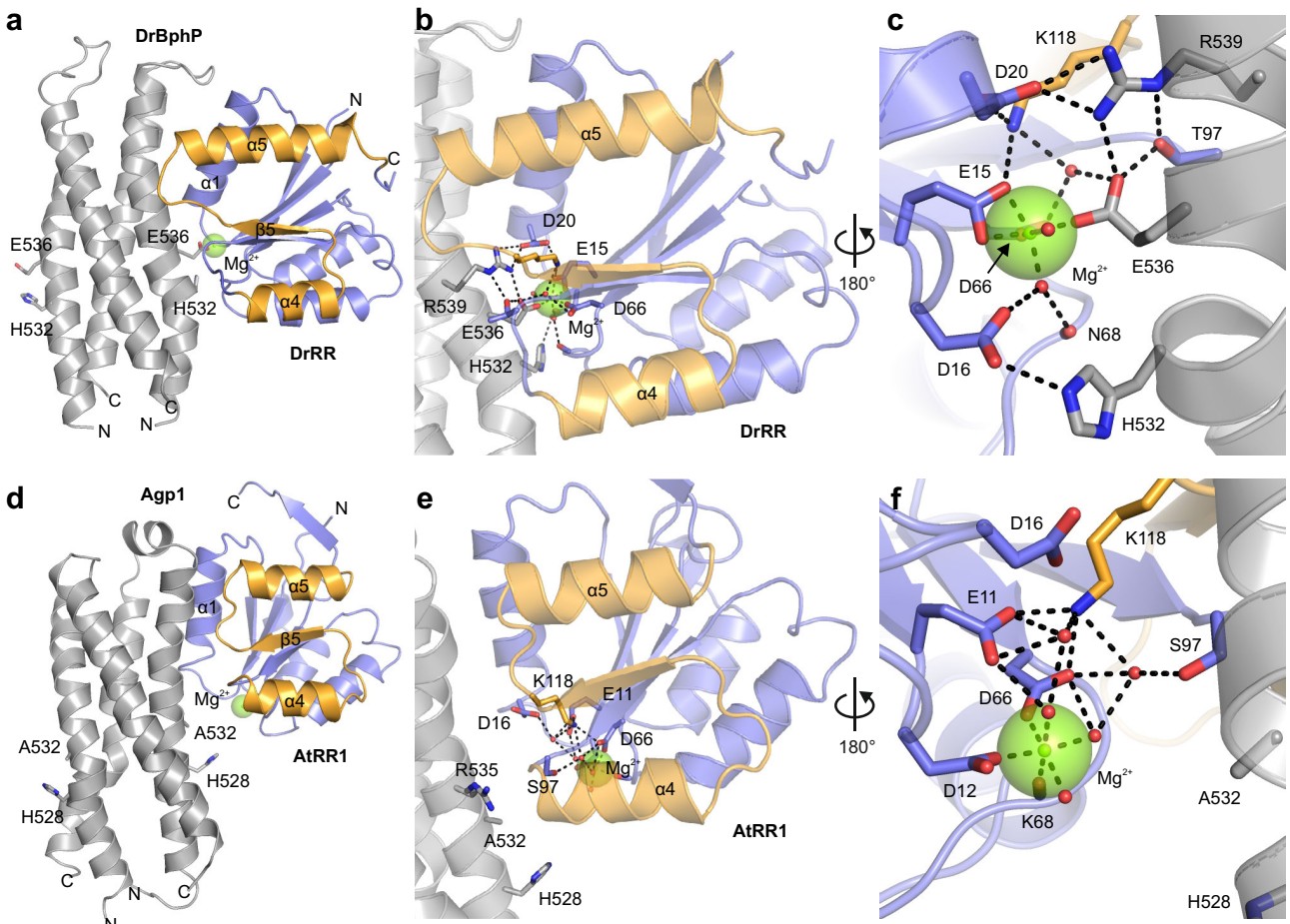

**Fig. 5 Complex models of the response regulators DrRR and AtRR1 and their interacting DHp domains. a–c** Model of the DrBphP/DrRR complex. **d–f** Model of the Agp1/AtRR1 complex. The overall structure of the complexes are shown in panels (**a**) and (**d**), detailed interactions around the active site in panels (**b**, **c**) and (**e**, **f**). The orientation of the complex is flipped by 180° in panels (**c**) and (**f**). Representative structural snapshots from the MD simulations are shown. The surrounding water box with Na$^+$ and Cl$^-$ ions is omitted for clarity, and only one monomer of the response regulator dimer is shown.

and well-defined salt bridges that are more pronounced in the *D. radiodurans* complex (Supplementary Fig. 6b, d). Notably, a set of interactions between DrBphP and DrRR, coordinated by a Mg$^{2+}$ ion (Fig. 5c), are absent in the Agp1/AtRR1 complex (Fig. 5f). We observe that in the DrBphP/DrRR complex, inter-chain salt bridges are less fluctuating in comparison to the Agp1/AtRR1 complex (Supplementary Fig. 6b, d). As a whole, the spatially confined and less stable interactions seen in the Agp1/AtRR1 model may account for the weak and transient binding observed experimentally for this complex (see Fig. 2 and Supplementary Figs. 1 and 2).

The DrBphP/DrRR complex model implies that Glu536 at the H + 4 position in the DHp domain coordinates with Mg$^{2+}$ and forms additional interactions with Arg539 and DrRR (Supplementary Fig. 6a, b). The corresponding residue in Agp1 is alanine (Ala532), and therefore these interactions are absent in the Agp1/AtRR1 complex model. DrBphP residue Arg539 forms a distinctive salt bridge with DrRR residue Asp20. The α5 helix is longer in DrRR than in AtRR1, which enables additional contacts between the β5–α5 loop and DrBphP. In our model, this positioning of the β5–α5 loop guides the side chain of Arg539 into close proximity to Asp20, thus enabling the salt bridge with DrRR (Fig. 5b, c). In the case of Agp1, the corresponding residue Arg535 points away from AtRR1 (Fig. 5e).

Taken together, the simulations implicate three central DrBphP residues that interact with the DrRR active site: His532, Glu536, and Arg539. These residues form a defined interaction network that includes a hexagonally coordinated Mg$^{2+}$ ion (Fig. 5c, Supplementary Fig. 6). We assessed the relevance of these residues for RR binding by ITC of selected DrBphP and Agp1 variants (Supplementary Fig. 2). In DrBphP E536A, DrRR interaction was only slightly reduced, consistent with the preservation of light-activated phosphatase activity in this variant (see Fig. 3d). Likewise, the corresponding A532E exchange in Agp1 did not notably affect the AtRR1 interaction (Supplementary Fig. 2e). These findings imply that the residue in the H + 4 position does not play a substantial role in the complex formation. By contrast, the arginine at the H + 7 position appeared important for the DrBphP/DrRR interaction, as its replacement by alanine abrogated binding (Supplementary Fig. 2d), as also reflected in a reduced phosphatase activity of the R539A variant (Supplementary Fig. 4d).

## Discussion

**The bacteriophytochrome from *D. radiodurans* is a light-activated phosphatase.** Bacterial phytochromes commonly act as light-regulated histidine kinases in two-component systems[32]. Here, we introduce biochemical and structural insight into the

activity of these phytochromes and their interaction with response regulators.

Agp1 acts as a red light-repressed histidine kinase that phosphorylates its cognate response regulator AtRR1[36] and that from *D. radiodurans* (Fig. 3a, b). Similar cross-reactivity has been reported for a bacteriophytochrome from *Pseudomonas syringae* (PsBphP) which can also phosphorylate DrRR[24]. Therefore, it is possible that Agp1 acts promiscuously and phosphorylates other response regulators in bacteria. By contrast, DrBphP did not show any kinase activity but functions exclusively as a phosphatase for DrRR (Fig. 3, Supplementary Figs. 3-4). Therefore, we assume that inside bacteria DrRR is phosphorylated non-enzymatically or by other histidine kinases in the cell. DrBphP can in turn dephosphorylate phospho-DrRR upon red-light exposure which likely triggers physiological responses. Notably, the precise enzymatic activity of DrBphP has been debated ever since its role in the control of carotene production was reported[21]. Our results now settle this long-standing debate and show that DrBphP is a biochemically active protein that dephosphorylates the DrRR, rather than phosphorylating it.

The fusion of the DrBphP PSM to the histidine kinase effector of Agp1 (Fig. 1a) produced a functional histidine kinase chimera (Fig. 3a, c), which shows that the DrBphP PSM is principally capable of controlling both histidine kinase and phosphatase activities in dependence of light. Consistent with this observation, the well-studied conformational changes of the DrBphP PSM during the Pr-to-Pfr transition[27,29,58] are similar across various phytochromes[41]. These findings indicate that both histidine kinase and phosphatase modules can be generally controlled by various phytochrome photosensory modules.

Even prior to the present elucidation of the enzymatic activity of DrBphP, its PSM has provided a versatile building block for light-controllable enzymes to be used in optogenetics. Pertinent enzymes have for instance been constructed through fusion of the DrBphP PSM with a cyclic-mononucleotide phosphodiesterase[59,60], a guanylate/adenylate cyclase[61,62], and a tyrosine kinase[63]. Our study introduces a red light-regulated HK chimera and phosphatase as a potential addition to the optogenetic toolkit.

**The binding modes of Agp1 and DrBphP support different functionalities**. Phytochrome photoactivation entails large-scale structural changes in the photosensory module, which are then relayed to the output module[27,29,58]. Although the molecular details of receptor activation are under debate and may differ between receptors, the conformational changes in the DHp bundle likely include rotation, bending, or changes in register of the constituent helices[18]. These conformational transitions can then change the interactions and/or enzymatic activity of the output HK domain[64].

The modes of binding to their cognate response regulators differ between DrBphP and Agp1, which may be integral to their respective activity profile. This difference manifested in dark reversion (Fig. 1b), in SEC profiles, in SPR and in ITC analyses (Fig. 2, Supplementary Figs. 1-2). The binding of AtRR1 to Agp1 was weak and transient, but the binding of DrBphP to DrRR had moderate affinity ($K_D \sim 10 \, \mu M$) and slower association/dissociation kinetics. Our structural data and models indicate that the binding interfaces in both complexes are generally similar but differ in their details: The DrBphP/DrRR complex had relatively stable interactions, whereas the interactions appeared transient and less defined in the Agp1/AtRR1 complex (Fig. 5).

We did not detect clear light-induced affinity changes in the BphP/RR pairs. These data concur with structural evidence that the RR binds to the DHp domain in a similar way regardless of whether the receptor resides in the kinase-active or phosphatase-active state.

Notably, the relevant interaction epitope of the DHp domain experiences only minor structural changes upon HK (in) activation[17,18]. The CA domain on the other hand binds to different DHp regions in darkness and upon light activation. Light-induced change in kinase activity may thus result either from different CA binding, varied accessibility of the catalytic histidine[17], or both. We note that the binding sites of the CA domain and RR partially overlap, as also manifest in the covariance analyses of the interfaces (Supplementary Fig. 7), which could create competition between the two binding schemes.

The structural changes in the DHp domain upon light activation may facilitate the switch between CA binding during the autokinase reaction and RR binding during the phospho-transfer and phosphatase reaction. As the RR competes with the CA domain for binding to the DHp domain, transient interactions between the molecules would be favored in the kinase-active receptor state. Structural asymmetry, observed for several HKs in their kinase-active state, may also facilitate the alternating binding of CA and RR[4,18,65]. While the phosphatase reaction is greatly facilitated by the CA domain[46] and ATP (Supplementary Fig. 4a, f), the CA binding in the phosphatase-active state likely differs from that of the kinase-active state. This difference may underlie the relatively slow binding kinetics between the DHp bundle and the phosphate-presenting response regulator (Fig. 2c, Supplementary Fig. 2).

**Two residues in the DHp helix govern the HK activity**. Two residues within the DHp domain, at positions +1 and +4 relative to the active-site histidine, have been implicated as particularly important for the enzymatic activity in the HisKA family. First, the autophosphorylation reaction involves a nucleophilic attack by the histidine to the γ-phosphate of ATP. This is facilitated by an acidic residue (aspartate or glutamate) in the H + 1 position acting as a general base[12]. Second, a threonine or asparagine residue in the H + 4 position governs phosphatase activity, potentially coordinating a water molecule for nucleophilic attack[14]. There is no crosstalk between the residues, as the H + 1 position does not contribute to the phosphatase activity, and the H + 4 position is not required for the kinase activity[14].

A large-scale sequence analysis of histidine kinases shows that the acidic residue in the H + 1 position is strictly conserved among the HisKA family (Fig. 3g), underlining its importance. If this residue is mutated, the kinase activity is impaired, as indicated by our results on the D529H mutant of Agp1 and wild-type DrBphP (Fig. 3b). A histidine in the H + 1 position is very rare among the HK sequences (Supplementary Fig. 8). Although important, the activity of DrBphP could not be rescued only by introducing an aspartate to this H + 1 position (Fig. 3b, Supplementary Fig. 4c). In HisKA proteins, the acidic residue in the H + 1 position is accompanied by an asparagine in the N-box of the CA domain, which stabilizes the active HK conformation and participates in phosphoryl transfer from ATP to the catalytic histidine[43]. Indeed, this asparagine shows a high level of conservation within the HisKA family (Fig. 3g). Our model of the Agp1 HK supports this interaction between the H + 1 aspartate (Asp529) and the N-box asparagine (Asn637) (Fig. 6a). By contrast, this interaction is likely absent in DrBphP, as the corresponding residues are both histidines (His533 and His648) (Fig. 6a). Consistent with this observation, the sequence analysis indicates that when the H + 1 position is a histidine, the conservation of the N-box asparagine is lost (Supplementary Fig. 8a).

DrBphP features Glu536 in the H + 4 position, indicating that this residue is central for the phosphatase activity. As previous studies on the phosphatase activity in HisKA proteins have mainly concentrated on threonine and asparagine residues, not much is known about the role of glutamate at this position[14,66–69]. That

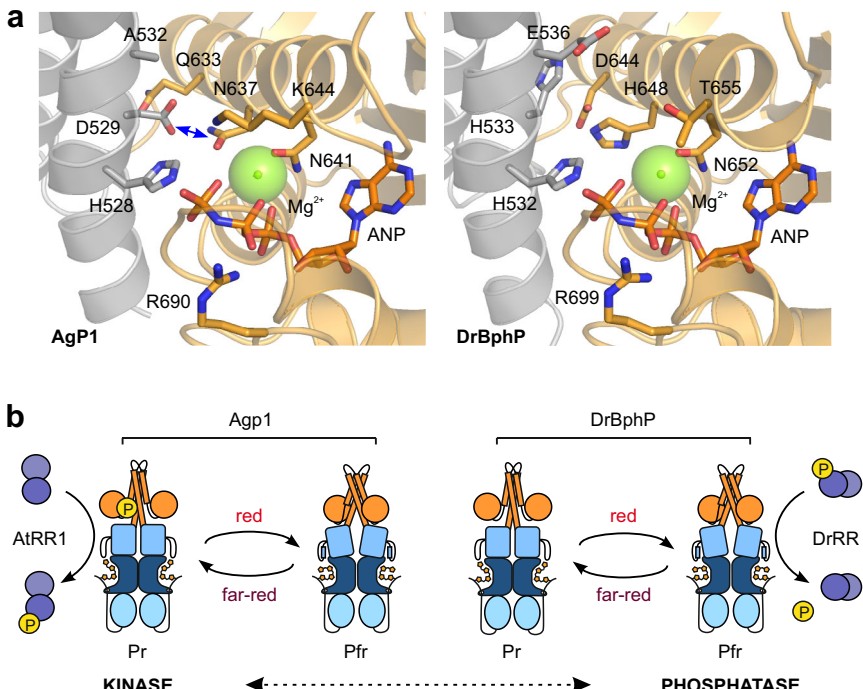

**Fig. 6 Structural comparison of residues important for the phosphotransfer reaction. a** Homology models of Agp1 and DrBphP based on the histidine kinase domain structure of HK853/EnvZ chimera in its phosphotransfer state (PDB 4KP4)[12]. The central residues are denoted as sticks, dimerization, and phosphohistidine (DHp) domain is shown in gray, and the catalytic ATP-binding (CA) domain is shown in orange. In Agp1, the potential interaction between His528 and Asn637 is shown as a blue arrow. **b** Proposed kinase and phosphatase activities of Agp1 and DrBphP. Many histidine kinases can function as both kinases and phosphatases, which may be governed by their DHp orientations[17]. In the case of the two phytochromes studied here, red light induces Pfr conformations that favor phosphatase activity. In the resting Pr state, the HK conformation favors kinase activity.

notwithstanding, glutamate is almost as conserved at this site as threonine or asparagine (Fig. 3g), which implies that all these residues play important roles, potentially fine-tuning the extent of the phosphatase reaction. This view is borne out by the variation in DrBphP phosphatase activity when testing different H + 4 substitutions (Fig. 3e). Changing this residue to alanine diminishes the phosphatase activity[14,66,68,69], like also demonstrated presently in the E536A variant of DrBphP and the wild-type Agp1 (Fig. 3d, f). However, conversely introducing the H + 4 glutamate in the Agp1 A532E mutation did not induce net phosphatase activity (Fig. 3f, Supplementary Fig. 4h), indicating that this position does not solely govern the Agp1 activity.

In our DrBphP/DrRR complex model, Glu536 forms a distinctive interaction with $Mg^{2+}$ in the active site (Fig. 5c). As this site would be occupied by the phosphate moiety of the phospho-DrRR, we consider it likely that Glu536 facilitates the dephosphorylation reaction. Notably, a glutamine at the H + 4 position retains activity, indicating that its side-chain amide group likely retains similar H-bond interactions as the carboxylate group of the glutamate (Fig. 3e). In addition to Glu536, Arg539 in the H + 7 position mediates the interactions at the DrRR active site (Fig. 5c, Supplementary Fig. 2d), suggesting a subsidiary role in the phosphatase reaction. This view is supported by the loss of conservation of both H + 7 arginine and H + 4 glutamate in HisKA proteins with an acidic residue at the H + 1 position (Supplementary Fig. 8a) and similar activity profiles of their alanine mutants (Supplementary Fig. 4d). In addition, the arginine at H + 7 position plays an important role in the interaction between DrBphP and DrRR (Fig. 5, Supplementary Fig. 2d).

**Model for bacteriophytochrome photoactivation**. To conclude, Asp529 in the H + 1 position of Agp1 is important for histidine kinase activity in the Pr state. In the case of DrBphP, His533 at H +

1 (along with other residues) renders the receptor inactive as a Pr-state kinase, whereas Glu536 at the H + 4 position makes it an effective phosphatase in the Pfr state (Fig. 3). We propose that the Pr conformation of bacteriophytochromes supplies a kinase-active state whereas the Pfr-like conformation prefers the phosphatase activity (Fig. 6b). Indeed, many HisKA family proteins function as both kinases and phosphatases, and these modes of action can be switched by a change of the relative orientation of DHp helices[17]. As demonstrated in the related histidine kinase YF1, blue light prompts quaternary transitions that channel into a register shift and super-coiling of the DHp helices[70,71]. Both kinase and phosphatase activities are therefore supported by conformational changes within the same structural framework. The bidirectional activity would also require both sets of activity-determining residues in the H + 1 and H + 4 positions of the HK domain, which seems to be the case in many phytochromes (Supplementary Fig. 8b).

Phytochrome function includes several structural tiers that range from the chromophore surroundings to large-scale structural changes in the entire protein as recently reviewed[72]. These tiers are in dynamic equilibrium, which can be shifted by the other tiers and by external factors[73]. The level of phytochrome output activity can be considered to be in an equilibrium between histidine kinase and phosphatase activities[18,65] (Fig. 6b). This equilibrium can be shifted to one direction by the light-induced changes in the photosensory module, and tuned by the sequence variation in the HK domain. In this study, we have shown how small differences in sequence dictate opposing enzymatic activities in two canonical phytochromes. In both cases, light controls their enzymatic activity.

## Methods

**Cloning and DNA material**. The phytochrome from *Deinococcus radiodurans* strain R1 (DrBphP, gene DR_A0050) in pET21b(+) plasmid (Novagen) was a kind gift from Prof. Richard Vierstra[21,74], and phytochrome 1 from *Agrobacterium*

*fabrum* strain C58 (Agp1, gene Atu1990) in pQE12 (Qiagen) was a kind gift from Prof. Tilman Lamparter[36]. Agp1 has a spontaneous R603C mutation, which resides on the surface of the CA domain. The mutations to DrBphP (H533D, E536A, E536T, E536N, E536D, E536Q, and R539A) and for Agp1 (D529H and A532E) were introduced with QuikChange Lightning Multi Site-Directed Mutagenesis Kit (Agilent Technologies). For cloning the chimera construct, DrBphP residues 513–755 were replaced with Agp1 residues 509–745. First, an XhoI restriction site was introduced after DrBphP residue 512 with QuikChange Lightning Multi Site-Directed Mutagenesis Kit (Agilent Technologies). Then, the C-terminal Agp1 fragment (aa 511–755) was ligated between the new XhoI site and an XhoI site right before the C-terminal His$_6$-tag. After introduction of the Agp1 fragment, the new XhoI site was changed to Agp1 residues 509–510 by site-directed mutagenesis. The response regulators from *Deinococcus radiodurans* strain R1 (DrRR, gene DR_A0049) and *Agrobacterium fabrum* strain C58 (AtRR1, gene Atu1989)[24] were produced as a service (Invitrogen). The response regulator constructs were cloned into pET21b(+) vectors (Novagen) by using restriction sites BamHI and XhoI. The EGFP-RR constructs were prepared with Gibson assembly cloning, in which N-terminal T7 tag of pET21b(+) was replaced with an EGFP-C1 sequence[75]. In addition, a linker of 10 residues (DSAGSAGSAG) was introduced with primers between the RR and EGFP sequences. For complete list of primers, see Supplementary Table 1.

**Sample expression and purification**. All DrBphP variants and the response regulators were expressed in *Escherichia coli* strain BL21 (DE3) overnight at 20–24 °C. After cell lysis with EmulsiFlex®, a molar excess of biliverdin hydrochloride (Frontier Scientific) was added to the phytochrome samples and incubated overnight on ice. No external biliverdin was added to the cell lysate in response regulator purifications. The His$_6$-tagged proteins were purified with NiNTA affinity purification using HisTrap™ columns (GE Healthcare), followed by size-exclusion chromatography (HiLoad™ 26/600 Superdex™ 200 pg, GE Healthcare) in buffer (30 mM Tris, pH 8.0)[76]. Agp1 and its D529H mutant were expressed in NEB Express® I$^q$ *E. coli* strain (New England Biolabs). The purification protocol was identical to other samples with a couple of exceptions: Protease inhibitor mix (ROCHE) and 0.5 mM TCEP were included in the sample before lysis, and affinity purification was conducted in (30 mM Tris/HCl, 150 mM NaCl, 1 mM TCEP) and varying imidazole concentration (5–500 mM). All purified protein samples were concentrated to 25–30 mg/ml in (30 mM Tris/HCl, pH 8.0) and flash-frozen.

**Absorption spectroscopy**. The dark reversion of the phytochromes was measured by the absorption spectroscopy using Agilent Cary 8454 UV-Visible spectrophotometer (Agilent). Absorption spectra in the wavelength range 690–850 nm were recorded from the mixture of response regulator and Pfr-populated BphP sample. The BphP samples were first diluted to 1.0 μM in (25 mm Tris/HCl, pH 7.8, 5 mM MgCl$_2$, 4 mM 2-mercaptoethanol, 5% ethylene glycol) to obtain an approximate A$_{700}$ value of 0.1 cm$^{-1}$. Ten times concentration (100 μM) of cognate response regulator was added into the BphP sample. Then the phytochromes were driven to a maximum population of the Pfr state by saturating illumination with 665 nm LED for 3 min, followed by immediate data acquisition in dark. Dark reversion data were recorded at 1 min intervals for the first 10 min, which was followed by intervals of 5 min up to 1 h and finally 10 min intervals until 2 h. All measurements were performed in dark at ambient conditions (room temperature). The steady-state spectra of Pr- and Pfr-state samples, in presence or in absence of cognate response regulator, were measured in the same buffer as for dark reversion. Pr state spectra were measured from the dark-adapted samples while Pfr spectra were measured after 3 min illumination with 665 nm LED.

The exponential fits from dark reversion data were calculated with Matlab R2019b (9.7.0.1190202) (MathWorks Inc.) using Eq. (1). In the case of DrBphP samples, three components were used for fitting, whereas two components were adequate for the rest of the samples.

$$\frac{A_{750}}{A_{700}}(t) = A_1 e^{-t/\tau_1} + A_2 e^{-t/\tau_2} + A_3 e^{-t/\tau_3} \qquad (1)$$

where $t$ is time, $A_{700}$ and $A_{750}$ are absorption values at specified wavelength, $A_n$ is the decay amplitude of the absorbance-ratio, and $\tau_n$ the time constant of the decay component.

**Size-exclusion chromatography (SEC)**. Size-exclusion chromatography with Superdex-200 Increase 3.2/300 (GE Healthcare) was conducted in the buffer (25 mM Tris/HCl pH 7.8, 5 mM MgCl$_2$, 4 mM 2-mercaptoethanol, 5% ethylene glycol). The absorption of proteins was detected at 489 and 280 nm. The illuminated samples (R) were pre-illuminated with 655 nm LED light for 5 min before injection. For each run, 24 μl of sample mixture (5 mg/ml each) was injected and eluted at 70 μl/min. The molecular weight estimates were determined by calculating a standard curve of marker proteins Vitamin B12 (1.35 kDa) myoglobin (17 kDa), ovalbumin (44 kDa), γ-globulin (158 kDa), and thyroglobulin (670 kDa).

**Surface plasmon resonance (SPR)**. For surface plasmon resonance measurements, phytochrome samples were dialyzed overnight to (20 mM HEPES, 300 mM NaCl$_2$, 5 mM MgCl$_2$, 0.10% (v/v) Tween20, pH 7.5) with a Spectra/Por® Micro

Float-A-Lyzer Dialysis Device (Spectrum Laboratories, USA). The measurements performed using Biacore X instrument (GE Healthcare). Response regulators were coupled onto carboxymethyldextran hydrogel-coated SPR Sensorchip (XanTec bioanalytics GmbH) according to manufacturer instructions. Each response regulator was coupled onto chip surface as 3 mg/mL (150 μM) in an acetate buffer (20 mM sodium acetate, pH 4.2) using EDC/NHS coupling protocol. The remaining activated groups on the sensor chip were then quenched with (1 M ethanol-amine-HCl, pH 8.5). The measurements were conducted by injecting 40 μL phytochrome sample at 20 μL/min, followed by wash step with (20 mM HEPES, 300 mM NaCl$_2$, 5 mM MgCl$_2$, 0.10% (v/v) Tween20, pH 7.5). Samples were either pre-illuminated with far-red (785 nm) or red (655 nm) LED light before injection, and all measurements were done in darkness at room temperature.

The sensorgrams were analyzed using the BIAevaluation-software version 4.1 (Biacore Life Sciences). The sharp peaks corresponding to the injection start (0 s) and stop (120 s) in each sensorgram were excluded from the analysis. For kinetic fit, a simple 1:1 interaction model between analyte and immobilized ligand was applied, followed by simultaneous fit of $k_a/k_d$ kinetics. The model is equivalent to the Langmuir isotherm for absorption to a surface. Steady-state binding levels ($R_{eq}$) were obtained by fitting a horizontal straight line to a chosen section of the sensorgrams (blue lines in Fig. 2c) and determining the average response. $R_{eq}$ values (y) and concentrations (x) were plotted in Origin 2018b and a nonlinear simple fit was obtained using the following Eq. (2) where A stands for concentration at $R_{eq}$.

$$R_{eq} = (A)R_{max} = (A) + K_D \qquad (2)$$

**Isothermal calorimetry (ITC)**. Isothermal calorimetry was conducted with MicroCal PEAQ-ITC (Mavern Pananalytical, United Kindom). For the measurements, the purified protein (in 30 mM Tris/HCl pH 8.0) were diluted 1:1 with 2× (50 mM Tris/HCl pH 7.8, 10 mM MgCl$_2$, 8 mM 2-mercaptoethanol, 10% ethylene glycol). BphP (30–50 μM, 300 μL) was loaded in the sample cell and RR (750–800 μM, 75 μL) was loaded in the injection syringe. To verify the Pr state of the BphP samples, they were briefly illuminated with 785 nm LED light just before sample application to the cell. The system was equilibrated to 25 °C with a stirring speed of 750 rpm in dark. Injection scheme started with a 0.4 μL response regulator injection, followed by 2 μL injections every 150 s. The ITC measurement (30 mM Tris/HCl, pH 8.0) were made using a Micro-200 ITC (MicroCal, Malvern). The concentrations used were 170–250 μM (BphP) and 750–800 μM (RR). BphP sample (206 μL) was loaded into the sample cell and RR (70 μL) was loaded into the injection syringe. The system was equilibrated to 25 °C with a stirring speed of 750 rpm. The injection scheme started with a 0.2 μL injection followed by 2 μL injections every 180 s. In both measurements, background signal was estimated by injection of response regulator into buffer and buffer into phytochrome with the same parameters. All data from triplicate experiments were analyzed using ORIGIN 7-based MicroCal PEAQ-ITC Analysis Software version 1.21 (Malvern Panalytical). The curves were fitted into a single-site binding isotherm with the first injection excluded. The $K_D$ value was reported as ±SD from three repeats.

**Radiolabeled kinase assay**. The radiolabeled kinase assay was done in a similar way to Lamparter et al.[36,77]. Purified BphPs and RRs were diluted to approximate concentrations of 3.5 μM (0.3 mg/ml) and 9 μM (1.7 mg/ml), respectively, in (25 mM Tris/HCl pH 7.8, 5 mM MgCl$_2$, 4 mM 2-mercaptoethanol, 5% ethylene glycol), and pre-illuminated briefly with saturating 785 nm LED light. Reaction was started by adding 3.7 kBq of [γ-$^{32}$P]ATP (PerkinElmer) in a total reaction volume of 10 μL. The samples were then incubated at 25 °C either in dark or under constant 655 nm LED illumination (5 mW/cm$^2$) for 20 min. The reaction was stopped by adding SDS sample buffer. The samples were then separated on 12% SDS-PAGE, and the gels were stained with Serva Blue, followed by drying in vacuum drier. The dry gels were then photographed and their radioactivity was monitored with an X-ray film. The experiment was repeated three times.

**Protein phosphorylation by acetyl phosphate and Phos-Tag detection**. In order to create phosphorylated response regulator we adapted the method described by McCleary and Stock[45]. There, response regulators (2–3 μg) were incubated with 50 mM acetyl phosphate for 30 min. The reactions were conducted at 37 °C in (25 mM Tris/HCl pH 7.8, 5 mM MgCl$_2$, 4 mM 2-mercaptoethanol, 5% ethylene glycol), followed by buffer exchange to (30 mM Tris/HCl, pH 8.0) with Vivaspin centrifugal concentrator (Sartorius, Germany). The final phosphoprotein concentrations were adjusted to 1.5 mg/ml (80 μM). Both kinase and phosphatase reactions were conducted in (25 mM Tris/HCl pH 7.8, 5 mM MgCl$_2$, 4 mM 2-mercaptoethanol, 5% ethylene glycol), where all the desired proteins (2–4 μg each) were incubated in 10 μl total volume at 25 °C, with or without 1 mM ATP. The reactions were started by adding ATP to the mixture and incubated either in dark or under saturating 657 nm red light. After 20–30 min, the reactions were stopped by adding 5× SDS loading buffer. For the mobility shift detection of phosphorylated RR proteins[35], we applied Zn$^{2+}$-Phos-tag® SDS-PAGE assay (Wako Chemicals). The 9% SDS-PAGE gels containing 25-μM Phos-tag acrylamide were prepared, and 10 μl of each reaction were run at 40 mA/gel at room temperature according to manufacturer instructions. See Source Data for full gels.

**Crystallography**. DrRR was crystallized with hanging drops vapor diffusion method. The protein of 10 mg/ml concentration was mixed in a 1:1 ratio with reservoir (0.1 M HEPES pH 7.5, 0.3 M $CaCl_2$, 25% PEG400). Crystals formed in few days and were flash-frozen in the reservoir solution containing 15% glycerol. The diffraction data were collected with 0.873 nm wavelength in beamline ID23-2 at the European Synchrotron Radiation Facility (ESRF). The data were processed with the XDS program package version on January 26, 2018[78]. The crystals belonged to space group P41212 with two dimers in an asymmetric unit. The initial phases were solved by molecular replacement with Phaser version 2.5.7[79]. As for a search model, a DrRR homology model was produced on-line with SWISS-MODEL workspace[80,81] and a crystal structure of a cyanobacterial response regulator RcpA (PDB code 1K68) as a template[50]. The structure was further refined with REFMAC version 5.8.0135[82] with automatic weighting and automatically generated local NCS restraints. The model building was done with Coot 0.8.2.[83]. For the final refinement cycles, six TLS regions for each protein chain were implemented from the TLS Motion Determination (version 1.4.0) web server[84]. The final structure had $R_{work}/R_{free}$ of 0.181/0.218. Statistics from data collection and refinement can be found in Table 1, and representative electron density of the final refinement can be found in Supplementary Fig. 9. Figures from crystal structures and complex models were created with the PyMOL Molecular Graphics System version 2.3.3 (Schrödinger, LLC).

**Computational modeling**. For computational simulations, DrBphP/DrRR and Agp1/AtRR1 complexes were constructed based on a crystal structure containing a sensor histidine kinase HK853 and its response regulator RR468 from *Thermotoga maritima* (PDB: 3DGE)[9]. Homology models consisting the dimeric DHp bundle of DrBphP (aa 520–592) and Agp1 (aa 513–584) were created on-line with SWISS-MODEL workspace (https://swissmodel.expasy.org/)[80,81] by using the corresponding DHp part of the *T. maritima* histidine kinase (aa 248–316) as a template structure[9]. As for the response regulators, the crystal structures AtRR1 response regulator (PDB: 5BRJ)[35] and DrRR (this paper) were applied as dimers. Waters that clashed with the interface and the phosphates at the active sites were not included in the models, whereas the $Ca^{2+}$ and $Mg^{2+}$ ion positions from the response regulator structures were retained and modeled with $Mg^{2+}$ ions.

To test whether the starting structure affects the modeled interactions, we repeated the modeling procedure as above to generate the DrBphP/DrRR and Agp1/AtRR1 complex models, but in this case used a crystal structure of a *T. maritima* ThkA/TrrA complex (PDB: 3A0R)[85] as a template.

Gromacs 2018.8[56] classical molecular dynamics package has been used to perform further modeling and simulations. Both Agp1/AtRR1 and DrBphP/DrRR complexes have been converted into the Gromacs topology, solvated within 15 × 15 × 15 nm periodic cubic box of water, and neutralized with counterions. In case of DrBphP/DrRR complex, we have replaced $Ca^{2+}$ ions which resides in DrRR crystal structure with $Mg^{2+}$ to be consistent with kinetic studies in solution. Additional $Na^+$ $Cl^-$ ions have been added to the neutralized cell in order to achieve 0.1 M total concentration of salt. Amber03[86] forcefield has been used for the proteins while water has been modeled with TIP3P[87] parameters.

Classical molecular dynamics simulations have been performed using the following protocol: at first we have minimized our systems for 10,000 steps with steepest descent method. Then 200 ns of Classical MD simulation has been performed within NPT ensemble at 300 K temperature using a V-rescale thermostat with 0.5 ps time constant[88] and at 1 atm. pressure using Parrinello-Rahman barostat with 1 ps time constant[89]. All bond lengths have been constrained to their equilibrium values, taken from the force field parameters with LINCS method[90], which allowed us to use a 2 fs time-step for the trajectory integration. A particle mesh Ewald (PME) method[91] has been used to account for periodic electrostatic interactions with real-space cutoff of 1.5 nm, while Lennard-Jones non-bonded interaction has been treated with a cut-off scheme using a range of 1.5 nm. RMSD of the backbone and RMSF of all the atoms of the proteins with respect to the initial configuration have been extracted from the trajectory.

Starting at 100 ns we have extracted snapshots each 10 ns and performed an extended analysis of interaction between kinases and response regulators. The interactions within the complex structures were analyzed with Protein interfaces, surfaces, and assemblies (PISA) service at the European Bioinformatics Institute (http://www.ebi.ac.uk/pdbe/prot_int/pistart.html)[57]. In addition, most prominent contacts have been analyzed by plotting contact distances throughout the MD simulation using the standard Gromacs trajectory tools.

Models for the DrBphP/DrRR and Agp1/AtRR1 complexes derived from both 3DGE and 3A0R structures, simulation parameters, and force fields used in the present work are available online at the GitHub repository: https://github.com/dmmorozo/HK-RR-simulations [https://doi.org/10.5281/zenodo.4922582]. In addition, the snapshots, extracted from the trajectories and further used in PISA analysis are also available online in the same repository.

**Sequence analysis**. To analyze sequence conservation and covariance in sensor histidine kinases, we conducted a BLAST (BLASTP version 2.10.0) search for the DHp and CA domains (residues 511–755) of DrBphP (Uniprot id BPHY_DEIRA, WP_010889310.1 [https://www.uniprot.org/uniprot/Q9RZA4]) against the non-redundant (nr) protein sequence database. Using the Biopython interface (version 1.77)[92] and custom Python (version 3.6.3) scripts[93], we retrieved the top

250,000 sequence hits, corresponding to an E-value cutoff of $5.0 \times 10^{-10}$. The sequences were clustered at a 30% identity level with UCLUST version 11.0.667[94], and the 11,994 cluster centroid sequences were determined. The original search sequence (WP_010889310.1 [https://www.uniprot.org/uniprot/Q9RZA4]) was added, and the sequences were aligned using MUSCLE (version 3.8.31)[95]. The consensus sequence of the alignment, mapped onto the search sequence, was plotted with WebLogo version 3.6[96]. Based on the alignment, covariance analysis was conducted with PSICOV (version 1.10D)[55] as described before[54]. Using custom Python scripts, the score matrix was plotted, and pairwise scores above a cutoff of 0.6 were mapped onto a homology model of the DHp and CA domains of DrBphP (Fig. 6a, Supplementary Fig. 7d). Homology models of DrBphP and Agp1 HK were calculated using SWISS-MODEL (https://swissmodel.expasy.org/)[97] based on the HK853/EnvZ chimera in its phosphotransfer state (PDB 4KP4)[12].

The sequence analysis of the response regulator proteins was carried out similarly. A BLAST search for the sequence of *D. radiodurans* RR (Q9RZA5_DEIRA, WP_010889309.1 [https://www.uniprot.org/uniprot/Q9RZA5]) provided 50,000 sequences with an E-value cutoff of $2.5 \times 10^{-10}$. Clustering at 50% identity yielded 4,338 sequences, to which were added those of the DrRR (WP_010889309.1 [https://www.uniprot.org/uniprot/Q9RZA5]) and AtRR1 proteins (Q7CY46_AGRFC, WP_121650967.1 [https://www.uniprot.org/uniprot/Q7CY46]). Sequence alignment and logo representation were done as for the histidine kinase data.

For the analysis of covariance between the histidine kinase and the RR (Supplementary Fig. 7), the above BLAST hits were scanned for proteins containing consecutive DHp, CA, and RR domains in a single polypetide chain. To be included in the subsequent analysis, entries were considered if they contained the Pfam HISKA, HISKA_2, or HISKA_3 domains[98], immediately followed by HATPase_c and Response_reg domains, with each domain not separated by more than 50 residues at maximum. The resultant 6,805 sequences were clustered at 50% identity, which left 5,386 centroid sequences. The amino acid sequences of the DrBphP (residues 511–755) and DrRR were concatenated and added. All sequences were then aligned as above and analyzed by PSICOV[55]. Pairwise scores above a cutoff of 0.6 were plotted onto a structural model of the DrBphP/DrRR complex (Supplementary Fig. 7c). The tabulated PSICOV scores are provided as a file Supplementary Data 1. In a control run, the aligned sequences were split into their DHp/CA and RR parts and randomly recombined before the analysis by PSICOV. The scrambling of the alignment abolished covariation between the DHp/CA and RR parts (Supplementary Fig. 7a, b).

**Reporting summary**. Further information on research design is available in the Nature Research Reporting Summary linked to this article.

## Data availability
The crystal data of DrRR generated in this study have been deposited in The Worldwide Protein Data Bank archive (http://www.wwpdb.org/) with accession code 6XVU. All published protein coordinates used in this study are also available in wwPDB under the accession codes 5BRJ, 3DGE, 4KP4, 1K68, 3DGE, 3A0R. The sequence datasets analyzed within this study are available in the Zenodo repository (https://doi.org/10.5281/zenodo.5005587). The sequence data used for sequence alignment is available in the UniProtKB (https://www.uniprot.org/) repository under the accession codes Q9RZA4, Q7CY45, Q097N3, Q09E27, Q6N5G3, Q6N5G2, F5Y2U7, Q885D3, Q9HWR3, B0JT05, Q55168, A9CI81, B9K3G4, Q1MCX7, B3PX96, A8HU76, Q6NB40, and A0A023X9Y5. The models generated and analyzed, simulation parameters, and force fields used in the current study are available in GitHub (https://github.com/dmmorozo/HK-RR-simulations) and Zenodo[99]. The authors declare that all relevant data supporting the findings of this study are available within the paper and its supplementary information files. The Source data file includes gels from representative experiments, and the gels covering the replicate experiments are available upon request from the corresponding author (Dr. Heikki Takala). Source data are provided with this paper.

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

## Acknowledgements

This work was supported by Academy of Finland grants 285461, 330678 (H.T.), 296135, and 332742 (J.A.I.), Jane and Aatos Erkko Foundation (J.A.I.), Three-year grant 2018–2020 from the University of Helsinki (E.M. and H.T.), and Bayreuth Humboldt Centre Senior Fellowship 2020 (E.M. and H.T.). S.W. and W.W. acknowledge the European Research Council for support (grant number: 279944), and B.S.-B. acknowledges Swiss National Science Foundation (P2ZHP2_164991). D.M. acknowledge the BioExcel CoE (www.bioexcel.eu), funded by the European Union contracts H2020-INFRAEDI-02-2018-823830, H2020-EINFRA-2015-1-675728. We acknowledge the European Synchrotron Radiation Facility (ESRF) for providing synchrotron access for crystal data collection and infrastructure support from the Biocenter Finland and the CSC-IT Finnish center for scientific computing for providing computational resources. We thank Dr. Harald Janovjak (Monash University) for advice on sequence analysis. We also thank M. Sc. Alli Liukkonen (University of Jyväskylä) for the assistance in laboratory, M.Sc. Moona Kurttila and Dr. Jessica Rumfeldt (University of Jyväskylä) for the help in the UV-vis data analysis, and Prof. Jari Ylänne (University of Jyväskylä) for the help with crystallography data and with radiolabeling assay, and Prof. Gerrit Groenhof (University of Jyväskylä) for the help with the MD simulations.

## Author contributions

H.T., A.M., and J.A.I. conceived the project. H.T., E.M., R.N., D.M., T.L., D.G., W.Y.W., and B.S.-B. conceived the experiments, and the results were analyzed together with J.R., V.P.H., S.W., J.A.I., and A.M. Paper was designed and written by H.T., A.M., J.A.I., and E.M. with input from all other authors.

## Competing interests

The authors declare no competing interests.
