## [Peer Review File · Nature Communications]

REVIEWER COMMENTS

Reviewer #1 (Remarks to the Author):

This manuscript studied the biochemical activities of bacterial phytochrome histidine kinases DrBphP and AgP1, and revealed their distinct enzymatic roles with AgP1 as a kinase and DrBphP as a phosphatase. The authors then used crystallography, sequence analyses and MD simulations trying to establish the structural features that account for different enzyme activities of the two proteins. Their distinct residues at H+1 and H+4 positions clearly contribute to the difference in enzyme activities, further adding evidences to the prior discovery of importance of these two residues. The manuscript ended the long puzzle about DrBphP regulation and demonstrated these bacterial phytochrome receptors as good model systems for TCS signaling. Using the chimera was a clever idea to investigate enzyme activity regulation. I found the results interesting and could be of general interest to understanding signaling mechanisms of two-component systems (TCSs). However, conclusions about the structure-function difference between AgP1 and DrBphP, especially those based solely on homology modeling of the HK-RR complex structure, could be strengthened with more validation experiments. My major and minor comments are summarized below:

Major points:

1. A significant amount of effort has been dedicated to the homology modeling and MD simulation results. I'm concerned about the accuracy of the modeled complex structures, especially for accuracy of side chain positioning that authors claimed to be responsible for the difference between AgP1 and DrBphP. Further, the modeled complex structures are heavily influenced by the initial HK853-RR468 structure. Will a different starting structure impact the simulation results? How homologous is AgP1 and DrBphP to HK853?

The modeling structures can certainly be valuable. But more experiments with mutants that potentially validate the authors' claims may further strengthen the manuscript. For example, the Arg residue at H+7 position is suggested to have different orientations in AgP1 and DrBphP, thus they may contribute differently to HK-RR interactions. Is it possible to mutate this Arg and assess the interaction to validate the structural prediction?

2. I'm not sure that covariance studies provide any new information. Data were presented with little interpretation. Are the results different from previous one studied by Laub and coworkers? Previous structural studies already showed that the interface residues between HK853-RR486 were consistent with covariance residues identified by Laub et al. It would be hard to imagine that structural models derived from HK853-RR486 would predict a set of interface contacts different from covariance residues.

3. In Figure 2A, is there any reason that AgP1 concentration did not go higher than 154 μ M in SPR studies? The current binding curve without saturation could not provide reliable estimation of affinity, one saturating data point at higher concentration, e.g. 267 μ M used for DrBphP, could greatly change the K_d value.

4. Page 5, line 164, it was suggested that AgP1/AtRR interaction was not affected by red light, but not all the data were consistent with this conclusion. To my opinion, SPR curves in Figure S2C did not indicate the absence of effect by red light, instead, the association/dissociation were too fast to draw conclusion. Actually, a slightly slower dissociation could be seen in S2C with red light than in dark conditions. Further, in Figure S1B, addition of AtRR (R) shifted the AgP1 peak to smaller retention volume thus higher molecular weight than with AtRR(D). Does this suggest stronger interaction in the presence of red light?

5. In Figure 3C & 3D, ATP was included to assess the phosphatase activity. Why not use ADP instead

of ATP? ATP is OK for DrBphP that does not display significant kinase activity. But for AgP1, ATP makes the reaction system complicated with simultaneous phosphatase and kinase reactions, thus phosphatase activity of AgP1 could not be evaluated. If ADP was used, it may allow assessment and comparison of phosphatase activities of both DrBphP and AgP1. AgP1 would be a good candidate for phosphatase studies for its unusual Ala residue at H+4 position, at which a polar or charged residue is usually expected to coordinate a water molecule for the phosphatase activity.

Minor points:

1. Different subfamilies of HKs have distinct mechanisms and structure features. It may help readers to associate with different mechanisms by stating early which subfamily do these two HKs belong to or the sequence features making them resemble a certain subfamily.
2. SEC data presentation in Figure S1 could be improved with more labeling and more interpretation in the legend. Molecular weight (MW) in parentheses is the actual MW of protein or value calculated from SEC? It may be a good idea to list their theoretical MW in the legend. Based on S1D, MW difference for DrRR and AtRR monomers would be ~8 kDa (26 kDa-37 kDa/2), but in S1B, the difference for EGFP fused RR is 45 kDa (90 kDa- 45 kDa). Why inconsistent? Did EGFP-DrRR behave differently from DrRR alone? DrRR were eluted as monomers in S1C and S1D, but EGFP-DrRR appeared as dimer or oligomer with multiple peaks? Will this impact the conclusion based on EGFP-RR?
3. Page 8, line 244, "did not unalter", typo for "did not alter"?
4. Sequence analyses in Figure 3F were based on 50,000 blast hits of DrBphP, only covering ~1/4 of HK sequences according Pfam counts. This is understandable for studying DrBphP. Would this be biased toward sequences similar to DrBphP that has an E at H+4? Once all HK sequences considered, will the E residue still be such frequently present?

Reviewer #2 (Remarks to the Author):

The manuscript 'Illuminating a Phytochrome Paradigm' by Elina Multamaki et al. investigates an enigma in the model bacteriophytochrome DrBphP from *Deinococcus radiodurans*. This light modulated enzyme lacks apparent histidine kinase activity, unlike related bacteriophytochroms, yet has an apparent function in control of carotene production.

By combining biochemical, structural and bioinformatics approaches in a comparative study involving DrbphP, a second phytochrome Agp1 and a chimeric protein the authors conclude the DrbphP serves as a light-controlled phosphatase, rather than a kinase. This is an important study. The manuscript has relevance to a number of fields, including those of signal transduction in general and of synthetic biology. The manuscript is also exceptionally well written and easy to follow.

I do have a small number of concerns that should be helpful in clarifying several issues to the reader of the manuscript:

Major

1. While the bacteriophytochrom histidine kinase proteins are relatively well described from a structural and signal detection level, the complete TCS's and their physiological roles are a little less clear. There are several times where this information would however seem essential for understanding. In particular:
 - a. What is the physiological role of the two systems studied here? I know it is mentioned in the text at some point in passing, but it would seem best to introduce it in the intro.
 - b. Are DrRR and AtRR both transcription factors and do they share homologous DNA-binding domains?

This seems important in the light of the structural characterization that suggest that they utilize different homodimerization modes. Ugguzoni et al. (PNAS 2017) showed that RR conserve different dimerization modes based on the choice of RR associated domain, in line with many previous structures.

c. The idea that DrBphP serves exclusively as a phosphatase for DrRR means that another kinase has to exist that phosphorylates DrRR. Alternatively, the phosphoryl group might originate from small phospho-donors in the cell such as acetyl phosphate. Is anything known about the origin of the phosphoryl group? Are there any orphan kinases in Dr? Would phosphorylation by acetyl phosphate make physiological sense? This should be discussed.

2. Much of the paper evolves around the notion that DrBphP has no kinase activity with potential explanations described on lines 203-214. While the evidence presented by the authors in their totality seems strong, it also seems plausible that proper kinase conditions have not been identified in vitro. I wonder if the authors or anyone else has ever explored alternatives to ATP as a co-substrate. Some histidine kinases have been shown to be GTP dependent. The observation that the DrRR protein crystalized with Calcium, rather than Mg in the active site also makes one wonder whether the DrBphP protein has a different metal preference. Has this been ruled out?

3. Regarding the Ca⁺⁺ ions in the DrRR structure, the reader is left to wonder if Ca⁺⁺ is the physiological metal ion utilized by this RR? This would seem to be in contrast to the Mg⁺⁺ ions that were used in the MD simulations. Is there a reason why the Ca⁺⁺ is an a crystallographic artifact? Please clarify.

Minor

4. Lines 49-50: 'To the extent it has been studied.... 2'. Reference 2 is 30 years old, yet the sentence describes the current state of knowledge. Perhaps a recent review might make for a better reference.

5. Line 53-54: DHP can be divided into four subtypes... What about Pfam family His_kinase? I believe that would be a fifth subtype.

6. Lines 84-85: the authors distinguish canonical and bathy phytochromes. I question if this is standard enough to where the casual reader does not need further explanation of what that means.

7. Page 9, Figure 3: There are a couple of things in this figure that are not immediately clear, although most are resolved much later in the text. These are: Why does AgP1 phosphorylate DrRR in in Fig 3B? How was total protein visualized? Was there ATP present in all conditions? In panel F, the relevance of the kinase and phosphatase residue choices is not clear here and only becomes apparent in the discussion. I feel all of this could confuse the non-expert reader and should be clarified in the figure legend and/or the nearby text.

8. Lines 320-321: The referenced manuscript 49 was an important study, but covariance analysis of HK/RR predates this study (e.g., White et al. 2007 Meth Enzymol; Burger et al. 2008 Mol Syst Biol.) and the statement that Laub pioneered covariance analysis is thus wrong. Subsequent approaches such as DCA and then PsiCov used here are also much more sophisticated and Laub and colleagues had no involvement in their development. Please adjust the text accordingly.

9. Lines 322-323: I am not entirely sure I understand the rational of utilizing hybrid receptors for HK-RR co-variance analysis. In the past, pairs were deduced by neighboring genomic context, which provides many more sequences. I assume that this was perhaps done because it was technically easier. If so, then please adjust the rational.

10. Figure 6B and legend: Is it correct that red light converts Pr to Pfr states and far-red light does the opposite? Or are the arrows reversed? Seems counter-intuitive.

Best,
Hendrik Szurmant

Reviewer #3 (Remarks to the Author):

The authors show that the RR of DrBphP interacts tightly with the phytochrome, unlike AtRR and AgP1, which interact weakly in light and dark. They determine that DrBphP is a light-regulated phosphatase, in contrast to AgP1 which is a light regulated kinase. A crystal structure of Dr RR shows it is a dimer and similar to some other dimeric phytochrome response regulators in particular that from the Cph1 pair, but that it does not contain the same unusual dimer interface of AtRR. Finally, the authors create refined models of kinase:RR pair structures for Dr and At phytochrome:RR based on a homologous structure from the EnvZ system, and they use these models to describe crucial amino acids important for the specific behaviours of the two systems.

The work is valuable, and everyone in the phytochrome field will take note of the light-regulated phosphatase results. The structural models of the protein complexes are interpreted in a way that is consistent with the data, but it feels like a stretch for Nature Communications to put so much faith in computational models to draw concrete conclusions about specific amino acid side chains.

This work will influence how researchers in the phytochrome field think about the biological implications of light on behaviour via two component signaling because of the entrenched thinking about kinase rather than phosphatase activity. It is less clear that the report will change the broader two component regulatory field, where researchers expect there are HKs with regulatable kinase or phosphatase activities. But, see #4 below -- if predictions are possible for other HKs this is more broadly relevant.

Comments:

1. References seem to be jumbled? Please very carefully check which reference is used in every case. For example, why is reference 23 (Arnold et al. SWISS MODEL) used to back up sentence, "Bacterial phytochromes usually belong to two-component signaling systems, with a cognate response regulator commonly encoded in the same operon."?
2. Perhaps along the same line, shouldn't reference 48 also be cited for the statement on lines 89-93 that "the dark adapted Pr state exhibited higher kinase activity than the Pfr state in Cph1, similar to other bacteriophytochromes ..."?
3. Binding affinities are measured using several techniques, which is to be commended. But what is referred to by the 1:1 stoichiometry? Phytochrome monomer: RR monomer? Phytochrome dimer: RR dimer? Phytochrome dimer: RR monomer?
4. The authors propose a predictive motif for kinases that are regulated kinases vs. regulated phosphatases. Are there other phytochromes that can be predicted to fall in the latter class based on this motif?
5. Figures use "D" and "R" presumably to refer to Dark and Red-illuminated forms? But I don't think these abbreviations are defined.
6. In final figures, please use vector graphics rather than a bitmap file in order to keep all of the text crisp.
7. How will the computational models of the phytochrome:RR complexes be made available to readers? They suggest testable hypotheses and their pdb files should be supplemental material.
8. Phos-tag has been used previously to detect phytochrome RR phosphorylation and could be cited in methods (ref. 48).

Reviewer #4 (Remarks to the Author):

The manuscript " Illuminating a Phytochrome Paradigm - a Light-Activated Phosphatase in Two-

Component Signaling Uncovered" by E. Multamäki and colleagues describes two canonical bacteriophytochromes (DrBphP and Agp1), which act as two-component signaling systems to transmit environmental stimuli to a response regulator. The authors propose the idea that AgP1 acts as a typical photosensitive histidine kinase (which has been shown before) in the two-component system, while identifying DrBphP as a photosensitive phosphatase (which is new here). Here, AgP1 can bind the corresponding response regulator only temporarily, DrBphP does so rather more strongly, which is supported by putative complex models. They suggest that two conserved residues have a strong influence on the balance between kinase and phosphatase activities, which has a direct effect on photoreception and two-component signaling.

The authors illuminate here a very difficult and controversial topic, since there are extremely fine-tuned mechanisms (most likely different somehow for various phytochromes), which probably depend on many factors, such as monomer/dimer constellations of phytochromes and RR, pH, ATP, light conditions and even enzymatic activities and functions.

The manuscript deals with a clever idea and is written very nicely. The result that DrBphP and AgP1 differ in their enzymatic activity and interactions is very interesting. In particular the dephosphorylation experiments of phospho-RR proteins are carried out very elegantly. The crystal structure analysis was also performed very solidly, as it can be estimated from the available data.

The work, which I appreciate very much, is on the one hand very well done, but on the other hand has a strong presumed part, with sometimes too little evidence. I have to say that without further biochemical data (mutations at the putative complex interface, other chimeras etc.) with subsequent interaction measurements such as SPR, I am somewhat reluctant to evaluate these complex models. The point is not that the MDs and the covariance analysis are well done, but that further biochemical evidence should be provided, which by the way was also done in the Skerker paper (Cell 2008). So the two complex models unfortunately remain somewhat ambiguous. The binding of AgP1 to AtRR is so different from the binding of DrBphP to DrRR that one can hardly believe it. I think that for this specific and important point, one can certainly make further elegant experiments to support the core findings of the manuscript. The question is also which role a dimer interface in RR could play here.

It is important to note that the authors should be careful not to say what the signaling is. It should be made clear to the reader whether it is the function of the protein (kinase or phosphatase function) or its physiological outcome (still unclear for both bacteriophytochromes).

Another question that stands out is whether there is not more than one RR for AgP1 (or even for DrBpHP), which may have an opposite effect, stronger and non-transient binding and phosphatase activity. Is it in principle possible that DrBphP cross-interact with other RR as shown for AgP1 with kinase activity? In addition, another issue is whether the bacteriophytochrome sample contains a part of the apo-protein in each case. And is this part similarly phosphorylated. The possible transfer of phosphate groups to such an apo-phytochrome to the responsible regulator was not investigated here. I miss also a short description of how the state of knowledge for plant phytochromes is, there is certainly more information than citation 36, which clearly defines the system in comparison to canonical bacteriophytochromes.

Several other remarks:

- The title "Illuminating a Phytochrome Paradigm – a Light-Activated Phosphatase in Two-Component Signaling Uncovered" is too imprecise for me, because it was only shown here for a bacteriophytochrome (DrBphP). I recommend to change it.
- Line 73/74: Is citation 23 here correct?
- Line 75/76: The PSM module has been proclaimed before, not only in citation 24. Perhaps one can

say, first described as a structure with an N-terminal photosensory module (PSM) divided into PAS (Period/ARNT/Single-minded), GAF (cGMP phosphodiesterase/adenylyl cyclase/FhlA) and PHY (Phytochrome-specific) domains.

- Line 81/82: The citation (27) is specific for cyanobacterial phytochromes, which do not contain a biliverdin. There are certainly other publications to cite.

- Why is the molecular weight in Fig.S1B for EGFP-labeled DrRR (90kDa) so much higher than for EGFP-AtRR (45kDa)? Are they either dimers or monomers? Isn't the possible monomer/dimer constellation (S1C/D) which is quite different in both RR in vitro also an important point to discuss? The question is also what significance monomers and dimers of the RRs might have in the kinase or phosphatase function of phytochromes.

- Were the SPR measurements performed completely in the dark? What was the apo-protein ratio to the bound protein species? Is anchoring to the chip a problem for the structural integrity of the protein?

- Line 183/184: The sentence is somewhat misleading. I can't find any experiments with ATP addition only with AMP-PNP. There are also no variations with Mg or a pH titration (only a buffer variation). Can the authors comment on this.

- Figure 3A: Autophosphorylation for AgP1 with and without RR was shown in Karniol and Viestra (PNAS 2003, Fig. 4C) for both Pr and Pfr states, with the Pr state showing only a slightly higher autophosphorylation. The experiments shown here clearly reveal a different picture without (or very low) autophosphorylation reaction in the Pfr. Why are there differences? The autophosphorylation (in the dark - Pr) of the chimera seems to be very low compared to the native Agp1. Is there a reason for this result? The phosphor-transfer activity of the chimera in the Pr state appears to work well in any case.

- The assay with phospho-DrRR is very nice as already mentioned. What is not quite clear to me yet is that the amount of phospho-DrRR in AgP1 and chimera is also reduced after red-light exposure in contrast to their dark state. Is the result not the same but only much weaker in DrBphP? It is not a question of doubting the results in this point, but I wonder if there is not also a low phosphatase activity in AgP1, which has a significance in the regulation of adjustable net kinase and phosphate activity? It would also be very exciting to see what would happen if a glutamate were incorporated into AgP1 instead of alanine as in DrBphP (here E536). Does it then have more net phosphatase activity? And what would occur if the chimeras turned around, i.e. the DrBphP output module in AgP1 is attached as a new chimera. Does it then also have an increased phosphatase activity after light exposure and no kinase activity (probably)? These experiments would further strengthen the facts.

- Is it in principle possible that DrBphP cross-interact with other RR as shown for Agp1 with kinase activity?

- There are many RSRZ outliers in the different monomers (C/D). Is the electron density for these monomers not so good (Are the B factors also higher than in A/B)?

- The sentences lines 406-410 are somewhat unclear, the physiological role is not yet fully understood and the results presented here rather show the phosphatase function of the protein itself, but not its physiological outcome.

REVIEWER COMMENTS

Reviewer #1:

This manuscript studied the biochemical activities of bacterial phytochrome histidine kinases DrBphP and AgP1, and revealed their distinct enzymatic roles with AgP1 as a kinase and DrBphP as a phosphatase. The authors then used crystallography, sequence analyses and MD simulations trying to establish the structural features that account for different enzyme activities of the two proteins. Their distinct residues at H+1 and H+4 positions clearly contribute to the difference in enzyme activities, further adding evidences to the prior discovery of importance of these two residues. The manuscript ended the long puzzle about DrBphP regulation and demonstrated these bacterial phytochrome receptors as good model systems for TCS signaling. Using the chimera was a clever idea to investigate enzyme activity regulation. I found the results interesting and could be of general interest to understanding signaling mechanisms of two-component systems (TCSs). However, conclusions about the structure-function difference between AgP1 and DrBphP, especially those based solely on homology modeling of the HK-RR complex structure, could be strengthened with more validation experiments. My major and minor comments are summarized below:

RESPONSE: We thank the reviewer for the positive opinion. By responding to the incisive comments, we believe to have improved the manuscript considerably.

Major points:

1. A significant amount of effort has been dedicated to the homology modeling and MD simulation results. I'm concerned about the accuracy of the modeled complex structures, especially for accuracy of side chain positioning that authors claimed to be responsible for the difference between AgP1 and DrBphP. Further, the modeled complex structures are heavily influenced by the initial HK853-RR468 structure. Will a different starting structure impact the simulation results? How homologous is AgP1 and DrBphP to HK853? The modeling structures can certainly be valuable. But more experiments with mutants that potentially validate the authors' claims may further strengthen the manuscript. For example, the Arg residue at H+7 position is suggested to have different orientations in AgP1 and DrBphP, thus they may contribute differently to HK-RR interactions. Is it possible to mutate this Arg and assess the interaction to validate the structural prediction?

RESPONSE: We thank the reviewer for raising the concerns about the modelled complex structures. Additional MD simulations and interaction assays with selected DrBphP and Agp1 mutants have now been carried out. We have included the results and discussed them in the manuscript text. In our view, this has strengthened the conclusions drawn from the homology modelling.

Currently there are crystal structures from three HK/RR complexes available: HK853/RR468 (PDB: 3DGE), DesK/DesR (PDB: 5IUN), and ThkA/TrrA (PDB: 3A0R). In all of these complexes, the core interactions are highly similar to each other and involve α 1 helices of both DHp and RR. In addition to this general interaction interface, each complex had auxiliary interactions that contribute to the specificity of the complex (for review see: Buschiazzo &

Trajtenberg (2019) *Annu. Rev. Microbiol.* 73, 507-528.). We assume that as far as the general interface interactions are reproduced, the flanking interactions at the RR active site should reflect the real situation. To clarify this, we have edited Figure 5 (panels B and E) to reflect the discussed features more accurately, and edited text accordingly.

To verify the 3DGE-based complexes presented in the manuscript, we created additional BphP/RR complex models that were based on the ThkA/TrrA complex structure (PDB: 3A0R). We added sections concerning these models to the Methods section and the Supplemental Information. The 3A0R-based DrBphP/DrRR complex structure behaved as a stable complex in MD simulations, which is visible in the backbone RMSD graph (Supporting Fig. S4H). Both 3DGE- and 3A0R-based DrBphP/DrRR complexes adopted highly similar interaction interfaces between the helices (Supporting Fig. S4F and G). This is also expected because of the high similarity of the features in the template crystal structures. The bending of the DHP helices, inherited from the 3A0R template, resulted in the catalytic His532 being further away from the DrRR active site. This bending might reflect a different activity state of the DHP bundle in DrBphP, but is caused by a complex-specific interaction in the template crystal structure. The 3A0R-based Agp1/AtRR1 complex was unstable in MD simulations and hence not considered further. To conclude, the different starting structures did not affect the main conclusions drawn from the 3DGE-based complex models.

To further verify the BphP/RR complex structures, we generated additional phytochrome mutants and tested with ITC whether their RR interactions are affected. These additional experiments are shown in Supporting Fig. S2C–E and include refined results for Fig. 2B. We discovered that DrBphP mutant E536A had only a slightly reduced affinity (K_D 13 μ M) to DrRR as compared to the wild-type counterpart (K_D 8 μ M). This small change indicates that the H+4 position does not play a prominent role in DrRR interaction. This was apparent in the H+4 mutant of Agp1 (A532E), which did not notably affect the Agp1/AtRR1 interaction. In addition, the H+4 interactions were affected by the template structure whereas the H+7 interactions were not (Supporting Fig. S4G). These additional experiments therefore support the presented BphP/RR models and their interactions *in vitro*.

As expected, the arginine in the H+7 position appeared to play an important role in the DrBphP/DrRR interaction. Its R539A mutation in DrBphP abrogated the interaction (Supporting Fig. S2D), which is consistent with the model predictions (Fig. 5C). This was further verified with MD simulations (Supporting Fig. S4I). There, the free energy change (ΔG) of HK-RR binding was ~18 kJ/mol higher in R539A than in the wild-type. This change corresponds to ~1000-fold higher K_D value for the R539A/DrRR interaction than for the wild-type DrBphP/DrRR interaction. This is fully consistent with the ITC data and supports the reviewer's hypothesis.

2. I'm not sure that covariance studies provide any new information. Data were presented with little interpretation. Are the results different from previous one studied by Laub and coworkers? Previous structural studies already showed that the interface residues between HK853-RR486 were consistent with covariance residues identified by Laub et al. It would be hard to imagine that structural models derived from HK853-RR486 would predict a set of interface contacts different from covariance residues.

RESPONSE: The chief motivation for conducting the analysis was to support the structural modeling and to pinpoint residue pairs in HK and RR with significant covariance values. The

overall results are in agreement with both Laub's analysis (Skerker *et al.* Cell 2007) and HK/RR complex structures. As noted by reviewer #2, more sensitive methods (such as "psicov" used here) became available since Laub's analysis. Moreover, Laub's data provided a graphical representation of the calculated covariances but no tabulated form, thus making it challenging to examine specific residues. We hence performed the covariance analysis using psicov. To allow the reader to inspect the results from the analysis in detail, we include them in tabulated form as supplementary material in the revised manuscript. That said, we concur with the reviewer that some more explanation was warranted and have changed the manuscript accordingly.

3. In Figure 2A, is there any reason that AgP1 concentration did not go higher than 154 μ M in SPR studies? The current binding curve without saturation could not provide reliable estimation of affinity, one saturating data point at higher concentration, e.g. 267 μ M used for DrBphP, could greatly change the K_D value.

RESPONSE: Unfortunately, there was an upper limit for the Agp1 concentration we could use in the experiment, and these experiments cannot be repeated with the same settings anymore. We however re-analyzed the data to evaluate the robustness of the results. We found that removing the highest concentration point in both analyses (Agp1+AtRR1 and DrBphP+DrRR) changed the K_D values by approximately 30%. Importantly, the main conclusion from the SPR study is unchanged, namely that binding of the AtRR1 by Agp1 is much weaker than for the corresponding *D. radiodurans* pair, which concurs with all other experiments.

To underline that the K_D value obtained from the Agp1 concentration series is of lower accuracy compared to DrBphP/DrRR, we added the following sentence in the Figure 2 legend: "As the Agp1 concentration levels did not reach saturation, the K_D estimation of the Agp1/AtRR1 interaction is less reliable than that of the DrBphP/DrRR pair."

4. Page 5, line 164, it was suggested that AgP1/AtRR interaction was not affected by red light, but not all the data were consistent with this conclusion. To my opinion, SPR curves in Figure S2C did not indicate the absence of effect by red light, instead, the association/dissociation were too fast to draw conclusion. Actually, a slightly slower dissociation could be seen in S2C with red light than in dark conditions.

RESPONSE: We thank the reviewer for this remark. Indeed, the SPR sensograms in the Supporting Fig. S2 (now in panel "I") do not rule out a slight effect of red light on binding. As for an effect on the dissociation rate mentioned by the reviewer, the possible small change is most probably due to baseline drifts that occurred during the measurement (distortion of the "R" curve at 20–200min). We have now changed the sentence in the main text according to the reviewer's view and allow for the possibility of a small effect of red light: "...the Agp1/AtRR1 interaction was not notably affected by red light" (Page 5, line 172).

Further, in Figure S1B, addition of AtRR (R) shifted the AgP1 peak to smaller retention volume thus higher molecular weight than with AtRR(D). Does this suggest stronger interaction in the presence of red light?

RESPONSE: According to our experiments, light does not affect the AtRR1 interactions with Agp1. In Supporting Fig. S1B, the shift of the Agp1 peak after red (R) illumination most likely results from aggregation of Agp1, not from binding of AtRR1. If there were interaction, the EGFP-AtRR1 retention would have been affected by the Agp1 addition. However, the EGFP-AtRR1 retention is unaffected by light, as seen in the lower right subpanel. The other SEC measurements presented in panels C–E do not indicate any AgP1/AtRR1 interaction either. Note that the experiment shown in S1C is new in the current version to complement the other SEC experiments.

5. In Figure 3C & 3D, ATP was included to assess the phosphatase activity. Why not use ADP instead of ATP? ATP is OK for DrBphP that does not display significant kinase activity. But for AgP1, ATP makes the reaction system complicated with simultaneous phosphatase and kinase reactions, thus phosphatase activity of AgP1 could not be evaluated. If ADP was used, it may allow assessment and comparison of phosphatase activities of both DrBphP and AgP1. AgP1 would be a good candidate for phosphatase studies for its unusual Ala residue at H+4 position, at which a polar or charged residue is usually expected to coordinate a water molecule for the phosphatase activity.

RESPONSE: We thank the reviewer for the comment, which stimulated us to do a series of new experiments. We have now tested the role of ADP in phosphatase reactions, presented in Supporting Fig. S3. Surprisingly, the phosphatase activity of DrBphP was reduced when ATP was replaced with ADP (panel F). Because of this unexpected result, we continued to include ATP when assessing DrBphP activity. We also show that Agp1 and its H+4 mutant A532E have very similar net activities with ATP (Fig. 3E). This indicates that the phosphatase activity of Agp1 is neither rescued nor enhanced solely by the H+4 residue.

We also attempted to assess the phosphatase activity of AgP1. As noted in the main text, acetyl phosphate was unable to phosphorylate AtRR1 efficiently (see Fig. 3B, well 8). For this reason, we could not efficiently produce phospho-AtRR1 for the phosphatase assays with ADP. Instead, we tested the phosphatase activity of the Agp1 variant with phospho-DrRR (Supporting Fig. S3H). This experiment showed that none of the Agp1 variants could dephosphorylate phospho-DrRR. This either means that there is no phosphatase cross-activity between Agp1 and DrRR or that Agp1 cannot function as a phosphatase altogether. Taken together, the new data provide additional insight into the phosphatase activity and its dependence on cofactors and residue exchanges in the BphP.

Minor points:

1. Different subfamilies of HKs have distinct mechanisms and structure features. It may help readers to associate with different mechanisms by stating early which subfamily do these two HKs belong to or the sequence features making them resemble a certain subfamily.

RESPONSE: We concur and provide this information now in the revised manuscript. Both Agp1 and DrBphP fall into the HisKA Pfam family. However, DrBphP is less closely related

to the Pfam HisKA consensus, yielding an E-value of around 10^{-3} , compared to an E-value of 4×10^{-9} for Agp1.

2. SEC data presentation in Figure S1 could be improved with more labeling and more interpretation in the legend. Molecular weight (MW) in parentheses is the actual MW of protein or value calculated from SEC? It may be a good idea to list their theoretical MW in the legend. Based on S1D, MW difference for DrRR and AtRR monomers would be ~8 kDa (26 kDa-37 kDa/2), but in S1B, the difference for EGFP fused RR is 45 kDa (90 kDa- 45 kDa). Why inconsistent? Did EGFP-DrRR behave differently from DrRR alone? DrRR were eluted as monomers in S1C and S1D, but EGFP-DrRR appeared as dimer or oligomer with multiple peaks? Will this impact the conclusion based on EGFP-RR?

RESPONSE: Following the reviewers suggestion, we have now added more labels and interpretation in the Supporting Fig. S1, as suggested. The monomeric molecular weights based on the amino acid sequence are now included in the figure legend, as well as additional text on the experimental settings and the interpretation of the results. The molecular weights in parentheses in the panels are calculated from the retention volumes, which is now indicated in the figure legend.

The HPLC measurements shown in Supporting Fig. S1B have some inconsistencies that were not found in other repeats (panels C–E). As the reviewer mentioned, the apparent sizes of EGFP-DrRR and EGFP-AtRR1 are dimers and monomers, respectively. This contrasts with the other SEC measurements and might be due to the harsh conditions in the HPLC column.

Now, we have included an additional SEC experiment as panel C, where the measurements were conducted in similar conditions as in B, but with FPLC and lower pressures. EGFP-DrRR appears monomeric, and EGFP-AtRR1 as a monomer/dimer mixture in this case. This is consistent with panels D and E. In addition, the change in EGFP-DrRR retention is clearly visible in both panels B and C, indicating that the conclusions about DrRR binding to DrBphP hold in both occasions.

3. Page 8, line 244, “did not unalter”, typo for “did not alter”?

RESPONSE: We thank the reviewer for pointing out this typo, which is now corrected. (Page 7, line 259)

4. Sequence analyses in Figure 3F were based on 50,000 blast hits of DrBphP, only covering ~1/4 of HK sequences according Pfam counts. This is understandable for studying DrBphP. Would this be biased toward sequences similar to DrBphP that has an E at H+4? Once all HK sequences considered, will the E residue still be such frequently present?

RESPONSE: The Pfam clan His_Kinase_A (CL0025) comprises both the DHP domain (denoted HisKA etc.) and the CA domain (denoted HATPase_c etc.). Out of the currently 496k members of this clan (Pfam v 33.1), around 200k correspond to DHP domains (families HisKA, HisKA_2, HisKA_3 and HWE_HK). To address this reviewer comment, we reran our BLAST

analysis and retrieved five times as many sequences as previously, i.e. 250k sequences. The subsequent analysis yielded similar results. As this reviewer may have suspected, the propensity for an E in the H+4 for HKs with an H in the H+1 position is slightly attenuated but still significantly above the value for all HKs.

Reviewer #2:

The manuscript ‘Illuminating a Phytochrome Paradigm’ by Elina Multamaki et al. investigates an enigma in the model bacteriophytochrome DrBphP from *Deinococcus radiodurans*. This light modulated enzyme lacks apparent histidine kinase activity, unlike related bacteriophytochroms, yet has an apparent function in control of carotene production.

By combining biochemical, structural and bioinformatics approaches in a comparative study involving DrbphP, a second phytochrome Agp1 and a chimeric protein the authors conclude the DrbphP serves as a light-controlled phosphatase, rather than a kinase. This is an important study. The manuscript has relevance to a number of fields, including those of signal transduction in general and of synthetic biology. The manuscript is also exceptionally well written and easy to follow.

I do have a small number of concerns that should be helpful in clarifying several issues to the reader of the manuscript:

RESPONSE: We are thankful for acknowledging the potential importance of the study.

Major

1. While the bacteriophytochrome histidine kinase proteins are relatively well described from a structural and signal detection level, the complete TCS's and their physiological roles are a little less clear. There are several times where this information would however seem essential for understanding. In particular:

a. What is the physiological role of the two systems studied here? I know it is mentioned in the text at some point in passing, but it would seem best to introduce it in the intro.

RESPONSE: We thank the referee for pointing out that the physiological roles of these systems indeed were not properly introduced. The original submission already contained information about the kinase activity of Agp1 (associated with bacterial conjugation). Now, we also include the proposed role of DrBphP by adding a sentence: “*DrBphP has been reported to play a role in the control of carotene production* (Davis et al. (1999) *Science*, 286: 2517-2520)” (Page 3, line 98)

b. Are DrRR and AtRR both transcription factors and do they share homologous DNA-binding domains? This seems important in the light of the structural characterization that suggest that they utilize different homodimerization modes. Ugguzoni et al. (PNAS 2017) showed that RR conserve different dimerization modes based on the choice of RR associated domain, in line with many previous structures.

RESPONSE: Both DrRR and AtRR1 consist only of a receiver (REC) domain. We have now included this information in the section on the DrRR crystal structure (Page 9). As these response regulators do not contain any DNA-binding domains, they cannot function as transcription factors *per se*. Therefore, the study by Ugguzoni et al. (2017) does not fully apply in this situation.

c. The idea that DrBphP serves exclusively as a phosphatase for DrRR means that another kinase has to exist that phosphorylates DrRR. Alternatively, the phosphoryl group might originate from small phospho-donors in the cell such as acetyl phosphate. Is anything known about the origin of the phosphoryl group? Are there any orphan kinases in Dr? Would phosphorylation by acetyl phosphate make physiological sense? This should be discussed.

RESPONSE: We agree with this reviewer's reasoning that inside *Deinococcus radiodurans* the DrRR might be phosphorylated by another SHK or low-weight compounds. Although little is known for *D. radiodurans* specifically, the phosphorylation of response regulators by high-energy phosphate donors, such as acetyl phosphate, is well established in many other bacteria (see doi: 10.1128/MMBR.69.1.12-50.2005). In light of these reports, we consider non-enzymatic phosphorylation of the DrRR inside its natural host organism possible.

Regarding a potential phosphorylation by other SHKs, according to Pfam *D. radiodurans* encodes 18 SHKs and 26 RR proteins. While testing the 17 SHKs other than DrBphP for their capability to phosphorylate DrRR is beyond the current scope, we analyzed the two groups of proteins for genomic proximity. Response regulators were found in genomic proximity for all but two SHKs. It is conceivable that these two "orphan" SHKs act on the DrRR, as proposed by this reviewer. That said, we note that these SHKs are both encoded on one of the two chromosomes of *D. radiodurans*, whereas DrBphP and DrRR are harbored in a so-called megaplasmid, which makes an interaction seem less likely.

2. Much of the paper revolves around the notion that DrBphP has no kinase activity with potential explanations described on lines 203-214. While the evidence presented by the authors in their totality seems strong, it also seems plausible that proper kinase conditions have not been identified in vitro. I wonder if the authors or anyone else has ever explored alternatives to ATP as a co-substrate. Some histidine kinases have been shown to be GTP dependent. The observation that the DrRR protein crystallized with Calcium, rather than Mg in the active site also makes one wonder whether the DrBphP protein has a different metal preference. Has this been ruled out?

RESPONSE: We thank the review for this insightful comment. We have now tested the DrBphP activity in additional conditions, see Supporting Fig. S3. We found out that the phosphatase activity is much reduced in the presence of ADP compared to ATP, and altogether inhibited when ATP is replaced by GTP (Supporting Fig. S3F). We think the packing

interactions in this crystal form require Ca^{2+} ions, but this does not mean that Ca^{2+} necessarily plays a role in the DrBphP activity. To address this question, we have now confirmed that replacement of Mg^{2+} by Ca^{2+} , incurs a loss of the DrBphP phosphatase activity (Supporting Fig. S3F). Histidine kinase activity was not detected for DrBphP in the presence of Ca^{2+} (Supporting Fig. S3G).

3. Regarding the Ca^{++} ions in the DrRR structure, the reader is left to wonder if Ca^{++} is the physiological metal ion utilized by this RR? This would seem to be in contrast to the Mg^{++} ions that were used in the MD simulations. Is there a reason why the Ca^{++} is an a crystallographic artifact? Please clarify.

RESPONSE: We thank the review for this comment. Like noted above, Ca^{2+} was required by the crystallization condition of DrRR. This is possibly due to a coordinated Ca^{2+} in the crystal packing interface (not shown), which may be hindered by the somewhat smaller size of the Mg^{2+} ion. As the figure below shows, our DrRR crystallization condition required CaCl_2 , which could not be successfully replaced with MgCl_2 .

Usually Mg^{2+} is deemed the physiological ion for two-component systems, but we now tested whether it can be replaced by Ca^{2+} . In Ca^{2+} -containing buffer the phosphatase activity of DrBphP was stalled and still no kinase activity was observable (Supporting Fig. S3F,G). Although enzymatic activity was lost, the binding between the DrBphP and DrRR was preserved (Supporting Fig. S2H).

Minor

4. Lines 49-50: ‘To the extent it has been studied..... 2’. Reference 2 is 30 years old, yet the sentence describes the current state of knowledge. Perhaps a recent review might make for a better reference.

RESPONSE: We agree and replace the reference with a newer one (Jacob-Dubuisson *et al. Nat Rev Micro* 2018). (Page 2, line 50)

5. Line 53-54: DHP can be divided into four subtypes... What about Pfam family His_kinase? I believe that would be a fifth subtype.

RESPONSE: We concur and adjust the text accordingly. Notably, the family His_kinase is not part of the His_Kinase_A clan, in contrast to the other four DHP families (HisKA, HisKA_2, HisKA_3 and HWE-HK). (Page 2, lines 53–56)

6. Lines 84-85: the authors distinguish canonical and bathy phytochromes. I question if this is standard enough to where the casual reader does not need further explanation of what that means.

RESPONSE: We agree and remove the reference to bathy phytochromes, which is not directly relevant to the manuscript.

7. Page 9, Figure 3: There are a couple of things in this figure that are not immediately clear, although most are resolved much later in the text. These are: Why does AgP1 phosphorylate DrRR in in Fig 3B? How was total protein visualized? Was there ATP present in all conditions? In panel F, the relevance of the kinase and phosphatase residue choices is not clear here and only becomes apparent in the discussion. I feel all of this could confuse the non-expert reader and should be clarified in the figure legend and/or the nearby text.

RESPONSE: We agree and in response to this comment, we have now included a more detailed explanation of the figure contents in the Figure 3 legend to assist the reader.

8. Lines 320-321: The referenced manuscript 49 was an important study, but covariance analysis of HK/RR predates this study (e.g., White et al. 2007 Meth Enzymol; Burger et al. 2008 Mol Syst Biol.) and the statement that Laub pioneered covariance analysis is thus wrong. Subsequent approaches such as DCA and then PsiCov used here are also much more sophisticated and Laub and colleagues had no involvement in their development. Please adjust the text accordingly.

RESPONSE: We thank the reviewer for drawing our attention to the earlier references, which we now cite in the revised manuscript. We apologize for the omission and rephrase the text accordingly. (Page 10, line 346)

9. Lines 322-323: I am not entirely sure I understand the rationale of utilizing hybrid receptors for HK-RR co-variance analysis. In the past, pairs were deduced by neighboring genomic context, which provides many more sequences. I assume that this was perhaps done because it was technically easier. If so, then please adjust the rationale.

RESPONSE: As this reviewer correctly notes, the covariance analysis requires the (correct) assignment of interacting HK/RR pairs. In the past, this assignment was commonly based on genomic context. Although genomic neighborhood generally hints at relevant interactions, there may be false positives that could affect the analysis. We opted for hybrid receptors as HK and RR form part of a single polypeptide chain, which allows the assignment of interacting pairs with enhanced confidence, as already explained in the original manuscript. The dataset used in our analysis comprised almost 5,400 sequences (after clustering at a level of maximally 50% sequence identity) which is sufficient for conducting the covariance analysis.

10. Figure 6B and legend: Is it correct that red light converts Pr to Pfr states and far-red light does the opposite? Or are the arrows reversed? Seems counter-intuitive.

RESPONSE: The arrows are the correct way around. The terms “Pr” and “Pfr” refer to the wavelength the phytochromes absorb. When they absorb the correct color of light (e.g. Pr absorbs red light), they photoswitch to a different state.

Reviewer #3:

The authors show that the RR of DrBphP interacts tightly with the phytochrome, unlike AtRR and AgP1, which interact weakly in light and dark. They determine that DrBphP is a light-regulated phosphatase, in contrast to AgP1 which is a light regulated kinase. A crystal structure of Dr RR shows it is a dimer and similar to some other dimeric phytochrome response regulators in particular that from the Cph1 pair, but that it does not contain the same unusual dimer interface of AtRR. Finally, the authors create refined models of kinase:RR pair structures for Dr and At phytochrome:RR based on a homologous structure from the EnvZ system, and they use these models to describe crucial amino acids important for the specific behaviours of the two systems.

The work is valuable, and everyone in the phytochrome field will take note of the light-regulated phosphatase results. The structural models of the protein complexes are interpreted in a way that is consistent with the data, but it feels like a stretch for Nature Communications to put so much faith in computational models to draw concrete conclusions about specific amino acid side chains.

This work will influence how researchers in the phytochrome field think about the biological implications of light on behaviour via two component signaling because of the entrenched thinking about kinase rather than phosphatase activity. It is less clear that the report will change the broader two component regulatory field, where researchers expect there are HKs with regulatable kinase or phosphatase activities. But, see #4 below -- if predictions are possible for other HKs this is more broadly relevant.

RESPONSE: We appreciate the positive review that acknowledges the potential importance of the manuscript. We have implemented several changes in response to the constructive comments and have thus improved the manuscript.

As for the predictive capability of our model, we can easily predict, but can unfortunately not verify the model since biochemical data is missing. However, we believe that our model will inspire more experiments on phytochromes and sensor histidine kinases to determine the biochemical activity, thus allowing the model to be tested in the future.

Comments:

1. References seem to be jumbled? Please very carefully check which reference is used in every case. For example, why is reference 23 (Arnold et al. SWISS MODEL) used to back up

sentence, “Bacterial phytochromes usually belong to two-component signaling systems, with a cognate response regulator commonly encoded in the same operon.”?

RESPONSE: The references have now been revised and corrected. The reference “Arnold et al. SWISS MODEL” refers now to “Auldridge and Forest (2011) *Crit. Rev. Biochem. Mol. Biol.* 46, 67-88.” (Page 2, line 79)

2. Perhaps along the same line, shouldn't reference 48 also be cited for the statement on lines 89-93 that “the dark adapted Pr state exhibited higher kinase activity than the Pfr state in Cph1, similar to other bacteriophytochromes ...”?

RESPONSE: The reference “Baker et al. (2016) *J. Bacteriol.* 198: 1218-1229.” has now been added, as suggested. During the revision, a colleague drew our attention to a relevant study on Cph1 kinase activity (Psakis *et al.* (2011) *Biochemistry.* 50, 6178-6188.) which we now cite as well. (Page 3, line 93)

3. Binding affinities are measured using several techniques, which is to be commended. But what is referred to by the 1:1 stoichiometry? Phytochrome monomer: RR monomer? Phytochrome dimer: RR dimer? Phytochrome dimer: RR monomer?

RESPONSE: Interesting point: Both phytochromes (DrBphP and Agp1) are homodimers in our conditions, whereas DrRR is monomeric and AtRR1 exists as a dimer/monomer mixture (Supporting Fig. S1). Due to the shifting monomer/dimer ratio of the response regulators, it is challenging to say which oligomeric state they assume during the titration. However, all molar concentrations used in the analyses refer to monomeric concentrations, and they are therefore stated as 1:1 molar binding models. We have changed the text accordingly (Page 5, line 164 and Page 6, line 193).

4. The authors propose a predictive motif for kinases that are regulated kinases vs. regulated phosphatases. Are there other phytochromes that can be predicted to fall in the latter class based on this motif?

REPOSENSE: In the Supporting Fig. S7, we have listed some of the well-known phytochromes with a histidine kinase effector domain. Most of the HisKA phytochromes contain features that would indicate both kinase (Asp at H+1) and phosphatase (Glu at H+4) activity. Phytochrome from *Ramlibacter tataouinensis* (RtBphP1) serves as an exception as it has Ala at H+4 position, like Agp1. This would indicate that both Agp1 and RtBphP1 have similar activity patterns, which is supported by Baker et al. (2016) *J. Bacteriol.* 198: 1218-1229. RtBphP1 has now been added to Supporting Fig. S7.

However, there is a general scarcity of experiments, which would characterize the HK/phosphatase activity of the phytochromes, and therefore the predictions cannot be verified. Bioinformatic searches would be suited to identify phytochromes that lack the sequence signatures of either the kinase or the phosphatase activities. A study of such proteins could serve as a test, and indeed represents an interesting question for future studies.

5. Figures use “D” and “R” presumably to refer to Dark and Red-illuminated forms? But I don’t think these abbreviations are defined.

RESPONSE: The letters “R” and “D” indeed mean red-illuminated and dark forms, respectively. We have now added this definition to relevant figures.

6. In final figures, please use vector graphics rather than a bitmap file in order to keep all of the text crisp.

RESPONSE: In the revised version, figures are supplied as vector graphics.

7. How will the computational models of the phytochrome:RR complexes be made available to readers? They suggest testable hypotheses and their pdb files should be supplemental material.

RESPONSE: All models used in the present work, along with the force fields, simulation parameter files and structures used in PISA analysis have been deposited and made openly available on the GitHub open repository: <https://github.com/dmmorozo/HK-RR-simulations>. We have now included the link to the repository and description under the Methods section.

8. Phos-tag has been used previously to detect phytochrome RR phosphorylation and could be cited in methods (ref. 48).

RESPONSE: We have now added the reference (Baker et al. (2016) *J. Bacteriol.* 198: 1218-1229) to the Methods section (Page 20, line 706).

Reviewer #4:

The manuscript " Illuminating a Phytochrome Paradigm - a Light-Activated Phosphatase in Two-Component Signaling Uncovered" by E. Multamäki and colleagues describes two canonical bacteriophytochromes (DrBphP and Agp1), which act as two-component signaling systems to transmit environmental stimuli to a response regulator. The authors propose the idea that AgP1 acts as a typical photosensitive histidine kinase (which has been shown before) in the two-component system, while identifying DrBphP as a photosensitive phosphatase (which is new here). Here, AgP1 can bind the corresponding response regulator only temporarily, DrBphP does so rather more strongly, which is supported by putative complex models. They suggest that two conserved residues have a strong influence on the balance between kinase and phosphatase activities, which has a direct effect on photoreception and two-component signaling.

The authors illuminate here a very difficult and controversial topic, since there are extremely fine-tuned mechanisms (most likely different somehow for various phytochromes), which probably depend on many factors, such as monomer/dimer constellations of phytochromes and RR, pH, ATP, light conditions and even enzymatic activities and functions.

The manuscript deals with a clever idea and is written very nicely. The result that DrBphP and AgP1 differ in their enzymatic activity and interactions is very interesting. In particular the dephosphorylation experiments of phospho-RR proteins are carried out very elegantly. The crystal structure analysis was also performed very solidly, as it can be estimated from the available data.

The work, which I appreciate very much, is on the one hand very well done, but on the other hand has a strong presumed part, with sometimes too little evidence. I have to say that without further biochemical data (mutations at the putative complex interface, other chimeras etc.) with subsequent interaction measurements such as SPR, I am somewhat reluctant to evaluate these complex models. The point is not that the MDs and the covariance analysis are well done, but that further biochemical evidence should be provided, which by the way was also done in the Skerker paper (Cell 2008). So the two complex models unfortunately remain somewhat ambiguous. The binding of AgP1 to AtRR is so different from the binding of DrBphP to DrRR that one can hardly believe it. I think that for this specific and important point, one can certainly make further elegant experiments to support the core findings of the manuscript. The question is also which role a dimer interface in RR could play here.

RESPONSE: We thank the reviewer for acknowledging the potential importance and broad relevance of the study. We have now addressed the raised concerns with additional interaction assays with selected DrBphP and Agp1 mutants and additional MD simulations. To summarize, we tested with ITC what effect H+4 and H+7 positions have on the BphP/RR complex formation (Supporting Fig. S2C–E). The results indicate that H+4 position does not affect the interaction much, whereas H+7 position is important for DrBphP/DrRR interaction. Additional MD simulations with a mutation in H+7 position verified the results (Supporting Fig. S4I), which are also consistent with our complex models (Fig. 5C).

Because similar concerns were also raised by another reviewer, please see the response to reviewer #1's major point 1 for additional details.

It is important to note that the authors should be careful not to say what the signaling is. It should be made clear to the reader whether it is the function of the protein (kinase or phosphatase function) or its physiological outcome (still unclear for both bacteriophytochromes).

RESPONSE: We agree, and have clarified the use of language in the paper. Specifically, throughout the manuscript we have studied the (enzymatic) activity of the model bacteriophytochromes, but not subsequent signaling events. We emphasized this in the text.

Another question that stands out is whether there is not more than one RR for AgP1 (or even for DrBpHP), which may have an opposite effect, stronger and non-transient binding and phosphatase activity. Is it in principle possible that DrBphP cross-interact with other RR as shown for AgP1 with kinase activity?

RESPONSE: This is an interesting proposal, which cannot be ruled out based on our data. For this study, we concentrated on the SHK/RR pairs that occur close to each other in the genome and that are reported in the previous literature. According to Pfam, *Agrobacterium fabrum*

encodes 63 response regulators and *Deinococcus radiodurans* encodes 26 response regulators. Therefore, it is in principle possible that Agp1 and DrBphP cross-interact with other RRs inside the cell. A comprehensive analysis of all possible interactions is beyond the scope of this study.

Directly pertaining to the reviewer comment, we have analyzed in more detail whether DrBphP can act on AtRR1 (see also the response to question 'c' by the Reviewer #2). We have conducted additional experiments, shown in Supporting Figures S2G and S3I, which indicate that DrBphP cannot cross-interact with AtRR1. Because acetyl phosphate phosphorylates AtRR1 poorly (Fig. 3B, well 8), measurements involving phospho-AtRR1 were not feasible. However, we tested whether DrBphP can de-phosphorylate phospho-AtRR1 in a competition experiment (Supporting Fig. S3I). There, both Agp1 (a kinase) and DrBphP (a phosphatase) were co-incubated with AtRR1 and ATP. As the amount of phosphorylated AtRR1 (created by Agp1) did not reduce when increasing the amount of DrBphP in the mixture, DrBphP did not seem to act as a phosphatase for AtRR1. The ITC measurements with mixed BphP/RR pairs (Supporting Fig. S2G) show that there was no clear DrBphP/AtRR1 or Agp1/DrRR interaction. The lack of DrBphP/AtRR1 interaction may explain the lack of the phosphatase cross-activity. However, although no Agp1/DrRR interaction was detected, Agp1 can still phosphorylate DrRR (Fig. 3B), which indicates that the histidine kinase activity does not require detectable interaction.

In addition, another issue is whether the bacteriophytochrome sample contains a part of the apo-protein in each case. And is this part similarly phosphorylated. The possible transfer of phosphate groups to such an apo-phytochrome to the responsible regulator was not investigated here.

RESPONSE: The biliverdin content of the BphP samples are routinely checked by inspecting the A700/A280 ratio of the phytochrome samples. This ratio provides a direct indication of the apoprotein content of the samples. Absorption spectra shown in the paper (Fig. 1B) indicate that the apoprotein content of our samples is very small.

We also tested the phosphatase activity of DrBphP apoprotein with the Phos-tag assay (Supporting Fig. S3F). The DrBphP apoprotein appeared like the Pr-state holoprotein in that it exhibited neither kinase nor strong phosphatase activities. In the case of Agp1, the apoprotein is shown to have strong histidine kinase activity (Karniol, B. & Vierstra (2003) PNAS, 100: 2807-2812.; Lamparter et al. (2002) PNAS, 99: 11628-11633.). Therefore, we conclude that the light-induced change in activity is specific to holoprotein. Even if there was a small amount of Agp1 or DrBphP apoprotein present, it would appear as a dark-state sample.

I miss also a short description of how the state of knowledge for plant phytochromes is, there is certainly more information than citation 36, which clearly defines the system in comparison to canonical bacteriophytochromes.

RESPONSE: We note that the reference (Golonka et al. (2019) Commun. Biol. 2, 448.) was not intended as a general source on plant phytochromes but was cited to support a specific observation in *A. thaliana* PhyB that contrasts with the present findings on bacterial phytochromes. To address this reviewer comment, we however added two general references to plant phytochromes in the introduction (Page 2, lines 75–77).

Several other remarks:

- The title “Illuminating a Phytochrome Paradigm – a Light-Activated Phosphatase in Two-Component Signaling Uncovered” is too imprecise for me, because it was only shown here for a bacteriophytochrome (DrBphP). I recommend to change it.

RESPONSE: We have now changed the title to “*Illuminating Two-Component Signaling with Model Bacteriophytochromes*”. This version should reflect the contents of the manuscript better.

- Line 73/74: Is citation 23 here correct?

RESPONSE: We thank the reviewer for pointing this error. The reference has now been corrected. (Page 2, line 79)

- Line 75/76: The PSM module has been proclaimed before, not only in citation 24. Perhaps one can say, first described as a structure with an N-terminal photosensory module (PSM) divided into PAS (Period/ARNT/Single-minded), GAF (cGMP phosphodiesterase/adenylyl cyclase/FhlA) and PHY (Phytochrome-specific) domains.

RESPONSE: We have modified the sentence according to the comment (Page 2, lines 79– 81).

- Line 81/82: The citation (27) is specific for cyanobacterial phytochromes, which do not contain a biliverdin. There are certainly other publications to cite.

RESPONSE: We agree that a citation to cyanobacterial phytochrome is not optimal here, and cite instead an article where biliverdin was identified for BphPs (Bhoo et al. (2001) *Nature*, 414:776-779.) (Page 3, line 86).

- Why is the molecular weight in Fig.S1B for EGFP-labeled DrRR (90kDa) so much higher than for EGFP-AtRR (45kDa)? Are they either dimers or monomers? Isn't the possible monomer/dimer constellation (S1C/D) which is quite different in both RR in vitro also an important point to discuss? The question is also what significance monomers and dimers of the RRs might have in the kinase or phosphatase function of phytochromes.

RESPONSE: Indeed, the HPLC measurements shown in Supporting Fig. S1B have some inconsistencies that were not found in other measurements. EGFP-DrRR appears larger than the EGFP-AtRR1, and potentially is a dimer. However, this does not apply in other SEC measurements (panels C–E), which were conducted in less harsh FPLC conditions.

We have now included more extensive description about the SEC measurements in the Supporting Fig. S1 legend. For more details, see response to the “minor point 2” from Reviewer #1.

Generally, it appears that the DrRR is monomeric and the AtRR mainly dimeric. At present, it is unclear which significance the oligomeric state has on kinase and phosphatase activities. More experiments are clearly warranted in the future to address this question.

- Were the SPR measurements performed completely in the dark? What was the apo-protein ratio to the bound protein species? Is anchoring to the chip a problem for the structural integrity of the protein?

RESPONSE: The SPR measurements were conducted in darkness, which is now indicated in the Figure 2 legend and specified in the Methods section. The apoprotein content of the DrBphP and Agp1 was non-significant, as indicated by the absorption spectra (A700/A280 ratio) of the samples. See the 4th response above for more details.

Anchoring of the DrRR and AtRR did not cause detectable problems with the integrity of these proteins. Although some of the RR molecules may orient themselves to the chip surface in such a way that does not support BphP interaction, we used excess of these proteins to counteract this effect. Although we cannot provide a direct measure, the general integrity is supported by 1) the observation of interactions, and 2) the qualitative agreement of the results with those from ITC.

- Line 183/184: The sentence is somewhat misleading. I can't find any experiments with ATP addition only with AMP-PNP. There are also no variations with Mg or a pH titration (only a buffer variation). Can the authors comment on this.

RESPONSE: We agree that Supplementary Figure S2B only shows the results with AMP-PNP. We therefore changed the sentence: "*Unlike in a blue light-regulated HK, this interaction was not affected by the addition of ATP analogue AMP-PNP (Supporting Fig. S2B).*" (Page 6, line194)

As for the Mg²⁺ and pH titrations, we did not titrate their effects in this study. We have however discovered that when Mg²⁺ is replaced with Ca²⁺, the DrBphP activity was lost (Supporting Fig. S3F) and the strength of the DrBphP/DrRR interaction is slightly changed (Supporting Fig. S2H).

- Figure 3A: Autophosphorylation for Agp1 with and without RR was shown in Karniol and Viestra (PNAS 2003, Fig. 4C) for both Pr and Pfr states, with the Pr state showing only a slightly higher autophosphorylation. The experiments shown here clearly reveal a different picture without (or very low) autophosphorylation reaction in the Pfr. Why are there differences? The autophosphorylation (in the dark - Pr) of the chimera seems to be very low compared to the native Agp1. Is there a reason for this result? The phosphor-transfer activity of the chimera in the Pr state appears to work well in any case.

RESPONSE: There are several potential reasons for the slight difference in the apparent Pfr-state activity of Agp1. These experiments have been conducted in different laboratories with different experimental details (e.g., illumination, apoprotein content, sample handling, exposure time of the gels). For example, different illumination conditions (time, wavelength, intensity) may result in different amounts of Pfr-state molecules. Also the apoprotein content

in the sample would appear as background activity in the red illuminated samples. Finally, the exposure time of the radiolabeled gels may cause saturation of the signals, which can overwhelm the differences between signal intensities. This variation in the exposure time is one of the reasons why Agp1 and Chimera differ slightly in Figure 4A. Generally, these proteins appeared to have highly comparable activity patterns. To conclude, regardless of the differences in the Agp1 activity between various papers, they are qualitatively consistent, and all the conclusions hold.

- The assay with phospho-DrRR is very nice as already mentioned. What is not quite clear to me yet is that the amount of phospho-DrRR in AgP1 and chimera is also reduced after red-light exposure in contrast to their dark state. Is the result not the same but only much weaker in DrBphP? It is not a question of doubting the results in this point, but I wonder if there is not also a low phosphatase activity in AgP1, which has a significance in the regulation of adjustable net kinase and phosphate activity? It would also be very exciting to see what would happen if a glutamate were incorporated into AgP1 instead of alanine as in DrBphP (here E536). Does it then have more net phosphatase activity? And what would occur if the chimeras turned around, i.e. the DrBphP output module in AgP1 is attached as a new chimera. Does it then also have an increased phosphatase activity after light exposure and no kinase activity (probably)? These experiments would further strengthen the facts.

RESPONSE: In the experiments, the amount of phospho-DrRR increases once Agp1 or Chimera are in the Pr state, indicative of their kinase activity. However, if the same experiment is repeated under red light, the amount of input phospho-DrRR does not reduce. This indicates that Agp1 and Chimera do not have net phosphatase activity towards phospho-DrRR. To verify if Agp1 has any phosphatase activity in the presence of ADP, or when A532 is mutated to E, we have conducted additional experiments, which are shown in Fig. 2E and Supporting Fig. S3. More discussion about these experiments can be found above in the reply to the major point 5 of the Reviewer #1.

Inspired by the suggestion, we have generated a reverse chimera that consists of the photosensory module of Agp1 and HK domain of DrBphP (AgPSM-DrHK). However, due to insolubility we could not purify it, preventing detailed studies.

- Is it in principle possible that DrBphP cross-interact with other RR as shown for Agp1 with kinase activity?

RESPONSE: We agree that inside of the bacterial cell cross-reactivity with other RRs is entirely possible, as demonstrated for other sensor histidine kinases. We provide a more detailed discussion of these aspects in the 3rd response above.

- There are many RSRZ outliers in the different monomers (C/D). Is the electron density for these monomers not so good (Are the B factors also higher than in A/B)?

RESPONSE: The overall electron density does not differ significantly between the DrRR monomers/chains. Poor density, and thus RSRZ outliers generally locate at the chain ends or sites that are not stabilized by crystal packing or other atoms (i.e., sites freely exposed to solution).

The asymmetric unit comprises of two dimers, made from chains A/B and C/D. Both A and D chains have 4 RSRZ outliers, whereas chains B and C have 8 outliers. The fact that chains C and D have more outliers can be explained by a loop region (residues 56–58), which contains glycines and is especially poorly resolved in these two monomers. The main-chain B-factors of these loop residues are generally higher than in the structure in average, being approximately 90–100 Å² (chain A), 110–130 Å² (chain B), 100–120 Å² (chain C), and 110–120 Å² (chain D).

- The sentences lines 406-410 are somewhat unclear, the physiological role is not yet fully understood and the results presented here rather show the phosphatase function of the protein itself, but not its physiological outcome.

RESPONSE: We agree that the physiological outcome of the DrBphP is not uncovered by our results, only the enzymatic function. We have edited the sentence “*and thereby elicit physiological responses*” (Page 13, lines 439–440) and hope that the difference between the enzymatic and the physiological activity is considered more clearly in the text.

REVIEWER COMMENTS

Reviewer #1 (Remarks to the Author):

The manuscript has been improved by revisions. Additional biochemical experiments and other revisions have addressed nearly all my concerns toward the original manuscript.

One of my main concern of the previous draft was the lack of biochemical evidence to support structural models and MD simulations. Although I still feel that it is a stretch to discuss amino acid side chain positions using a computational structural model, the newly added ITC and phosphorylation data on H+4 and H+7 mutants provided some validation of the modeled complex structures. It is implicated that H+4 and H+7 residues may contribute to a stronger HK-RR interaction and higher phosphatase activity for DrBphP than Agp1. However, E536A (H+4) showed minimal change in affinity, which appears not fully consistent with the model? Can this be explained by simulation? What would be the predicted $\Delta\Delta G$ of E536A?

Minor issues:

1. Page 5, line 169, "...($K_D > 200 \mu M$)" and Fig. S2J. I do agree with the conclusion that Agp1/AtRR has a weaker affinity based on all the data. But it may be better not to claim $K_D > 200 \mu M$ because the fitting is less reliable as mentioned in the legend. What are the confidence levels for estimated K_D values and how the fitting was done? The adj. R^2 for this fitting is 1.0 (Fig. S2J), suggesting that either the data were overfitted or the fitting might not be properly done. Moreover, adj. R^2 stands for adjusted R^2 , not adjacent R^2 as described in Fig. S2J legend.

2. page 5, Fig. 2C, ITC results, K_D , 10.5 ± 1.5 . This K_D value of $10.5 \mu M$ appeared to be derived from experiments performed in 25 mM Tris buffer, but the corresponding data were not shown in either Fig. 2B or S2. The ITC data in Fig. 2B were performed in 50 mM Tris buffer. The mismatch of two K_D values in Fig. 2B and 2C is confusing, why not change 10.5 in Fig. 2C to 8.1?

3. I find it generally difficult to follow all the SEC (Fig. S1) and ITC data (Fig. S2) because of multiple buffer conditions used. All these buffer conditions were included in legends but hard to find due to the length of legends. Is it possible to label each subgraph with the corresponding buffers and dark/light conditions?

Reviewer #3 (Remarks to the Author):

Three reviewers were concerned originally about the heavy reliance on models of the structural interactions between phytochromes and response regulators. The authors have expanded their simulations and have experimentally measured the interaction as well as biochemical activity in additional mutants. My original concern was partly alleviated. Much of the new data is included in the Supplemental Information due presumably to the space restrictions on article length, which is too bad.

Are hand-in-hand (main text) and arm-in-arm (supplemental info) the same thing?

Reviewer #4 (Remarks to the Author):

The authors have incorporated many of the reviewers' suggestions.

However, there are still critical points that need to be addressed.

The presented molecular dynamic computer simulations are still somewhat difficult. Root-mean-square deviation of atomic positions (RMSD) values for Agp1-AtRR1 in Fig. S4C of up to 4 Å are extremely high. The resulting statements in detail are then simply very vague, as reviewer 1 has already stated in a glass-clear manner. It is also really not better that the new simulations in Fig. S4H based on ThkA/TrrA complex structure (PDB ID:3A0R) resulted in even higher RMSD (up to 5 Å) and that this time also for the DrBphP/DrRR simulation (up to 2.5 Å). Besides, the 3A0R label in C and H appears as if the same model was used for both calculations. This is surely an error and should be improved. I can't see distances in F and G either, to really evaluate the potential contacts. I can't really understand the conclusions either. In both models, the conformations of H+4 (E536) and H+7 (R539) do drastically change (Fig. S4F/G). I do not like to say it but I would cut down the MD data as it hardly helps.

A key point for DrBphP is that Glu536 is central in the H+4 position, for the phosphatase activity argumentation as a key residue in DrBphP. The substitution to Ala is important, but also very heavy. I don't really understand why the authors don't explicitly analyze it in a bit more detail, even though they have addressed it themselves. I strongly suggest 2-3 more substitutions at the H+4 position to verify the argument, for example to Asp and to Gln or even to Thr or Asn (as in phosphatase activity in HisKA proteins). It would be very interesting and also important to show if the phosphatase activity can be somehow rescued. In general, the verification of the statements in the manuscript via mutational experiments for DrBphP is weak. There are only exactly three test mutations, but many important replacements that have not been made.

If the Agp1 concentration does not reach saturation (Fig.2), the KD value should probably not be reported. I also find it exceedingly difficult that the results of the follow-up experiments differ so much, which argues for not providing a KD determination, but simply saying that Agp1 binds weakly to At1RR1.

We thank the Reviewers for the careful study of our manuscript. As detailed below, we have further revised the manuscript in four principal ways:

1. We have performed substantial new experiments that address in more detail the relevance of key residues at the H+4 position for the phosphatase activity.
2. Upon studying the reviewers' comments, we have de-emphasized and partially removed the computational results.
3. Following your suggestion, we have rearranged the figures in the main manuscript and the supplemental materials.
4. We have carefully revised the entire text for enhanced clarity.

The changes in the manuscript text are indicated in blue colour, and the point-by-point responses to the Reviewer's comments can be found below. Taken together, we deem the manuscript further improved and hope that it has thus become suitable for publication in *Nature Communications*.

REVIEWER COMMENTS

Reviewer #1:

The manuscript has been improved by revisions. Additional biochemical experiments and other revisions have addressed nearly all my concerns toward the original manuscript.

One of my main concern of the previous draft was the lack of biochemical evidence to support structural models and MD simulations. Although I still feel that it is a stretch to discuss amino acid side chain positions using a computational structural model, the newly added ITC and phosphorylation data on H+4 and H+7 mutants provided some validation of the modeled complex structures. It is implicated that H+4 and H+7 residues may contribute to a stronger HK-RR interaction and higher phosphatase activity for DrBphP than Agp1. However, E536A (H+4) showed minimal change in affinity, which appears not fully consistent with the model? Can this be explained by simulation? What would be the predicted $\Delta\Delta G$ of E536A?

RESPONSE: We acknowledge this Reviewer's reservations about the computational modelling. Specifically, we concur that a reliable assessment of individual amino acids is challenging and may amount to overinterpretation of the computational results. As a case in point, $\Delta\Delta G$ calculations for the E536A substitution yielded rather differing values, depending on which complex model was used and whether Mg^{2+} or Na^+ ions were occupied at the active site. Albeit being mostly in line with the experimental results, we have removed the $\Delta\Delta G$ calculations entirely from the manuscript in light of this uncertainty. We have changed respective parts of the manuscript and toned down the interpretation of the computational results.

More generally, we note that the main merit of the simulations has been to motivate additional

experiments, which have notably improved the paper. At the same time, our interpretation and main conclusions of the manuscript are supported by the experimental data and not solely by the simulation.

Minor issues:

1. Page 5, line 169, “...(KD>200 uM)” and Fig. S2J. I do agree with the conclusion that Agp1/AtRR has a weaker affinity based on all the data. But it may be better not to claim KD>200 uM because the fitting is less reliable as mentioned in the legend. What are the confidence levels for estimated KD values and how the fitting was done? The adj. R2 for this fitting is 1.0 (Fig. S2J), suggesting that either the data were overfitted or the fitting might not be properly done. Moreover, adj. R2 stands for adjusted R2, not adjacent R2 as described in Fig. S2J legend.

RESPONSE: We concur that the statement that K_D is above 200 μM is not well substantiated by the data. Owing to the weak affinity, even at the highest achievable Agp1 concentration of 154 μM , the interaction could not be saturated. We hence decided to remove the fit for this interaction and instead estimate the affinity to be on the order of hundreds of micromolar.

We formulated the manuscript text as follows: “*Consistent with the SEC analysis, the interaction between Agp1 and AtRR1 was substantially weaker, and the binding curve did not reach saturation at the highest achievable Agp1 concentration of 154 μM . We hence estimated the affinity to be on the order of hundreds of micromolar.*”

We thank the Reviewer for pointing out the error in term *adj. R²*. This has now been corrected. Although now removed from the manuscript, the fit for the Agp1/AtRR1 had an unrounded adjusted R^2 value of 0.997. This indicates that the fit agreed very well with the available data points.

2. page 5, Fig. 2C, ITC results, K_D , 10.5 ± 1.5 . This K_D value of 10.5 μM appeared to be derived from experiments performed in 25 mM Tris buffer, but the corresponding data were not shown in either Fig. 2B or S2. The ITC data in Fig. 2B were performed in 50 mM Tris buffer. The mismatch of two K_D values in Fig. 2B and 2C is confusing, why not change 10.5 in Fig. 2C to 8.1?

RESPONSE: We thank the Reviewer for spotting this typo! We have now changed the value in panel 2C (now Supporting Figure S2I) to $(8.1 \pm 1.3) \mu\text{M}$.

3. I find it generally difficult to follow all the SEC (Fig. S1) and ITC data (Fig. S2) because of multiple buffer conditions used. All these buffer conditions were included in legends but hard to find due to the length of legends. Is it possible to label each subgraph with the corresponding buffers and dark/light conditions?

RESPONSE: We agree that the SEC data were not as clearly presented as possible. We have now added additional labels to the SEC panels in the Supporting Figure S1. In addition, we have also moved one panel (former S1C) to the main text and removed one redundant experiment (former S1E), which should clarify the SI figure. In ITC data (Supporting Figure S2), all measurements have been done in dark and the only buffer condition that differs from others is in panel A. We have now labelled this buffer condition in S2A.

Reviewer #3:

Three reviewers were concerned originally about the heavy reliance on models of the structural interactions between phytochromes and response regulators. The authors have expanded their simulations and have experimentally measured the interaction as well as biochemical activity in additional mutants. My original concern was partly alleviated. Much of the new data is included in the Supplemental Information due presumably to the space restrictions on article length, which is too bad.

RESPONSE: We thank the Reviewer for the suggestion and have now moved data from the supplemental material. These include panels S1C and S1E. In addition, we have rearranged panel position in S1 and S2, which has made the Supporting Figures more accessible.

Are hand-in-hand (main text) and arm-in-arm (supplemental info) the same thing?

RESPONSE: We have now unified the term as “*arm-in-arm*”, which is the one used in the referred paper.

Reviewer #4:

The authors have incorporated many of the reviewers' suggestions. However, there are still critical points that need to be addressed.

The presented molecular dynamic computer simulations are still somewhat difficult. Root-mean-square deviation of atomic positions (RMSD) values for Agp1-AtRR1 in Fig. S4C of up to 4 Å are extremely high. The resulting statements in detail are then simply very vague, as reviewer 1 has already stated in a glass-clear manner. It is also really not better that the new simulations in Fig. S4H based on ThkA/TrrA complex structure (PDB ID:3A0R) resulted in even higher RMSD (up to 5 Å) and that this time also for the DrBphP/DrRR simulation (up to 2.5 Å).

RESPONSE: We realize that we may have presented these results in an unclear manner. The root-mean square deviation of atomic positions (RMSD) shown now in Fig. S5C and H refer to the starting model. Given that the starting model is partially based on homology and individual crystals structures, an initial, more substantial adjustment occurred early in the simulation. By contrast, afterwards the fluctuation of the RMSD value was moderate (generally within 1Å). The small fluctuations therefore indicate that the complex models are relatively stable. To convey this aspect, we modified the plots in panels S5C and S5H to show the initial RMSD jump. In addition, we have modified the figure legend and text to clarify that the RMSD values depict the values relative to the starting structure.

As noted in our response to Reviewer #1, we also generally toned down the interpretation of the computational results.

Besides, the 3A0R label in C and H appears as if the same model was used for both calculations. This is surely an error and should be improved.

RESPONSE: We thank the Reviewer for pointing out the labelling error in panel S5C, which is now corrected. The starting structure was indeed based on the 3DGE-based model.

I can't see distances in F and G either, to really evaluate the potential contacts. I can't really understand the conclusions either. In both models, the conformations of H+4 (E536) and H+7 (R539) do drastically change (Fig. S4F/G). I do not like to say it but I would cut down the MD data as it hardly helps.

RESPONSE: We appreciate this comment, which is related to remarks by Reviewer #1. We concur that results obtained from simulations need to be interpreted with more care. In response, we have thus cut parts of the MD analysis (see above comments). On a general note, the simulations possess merit as they motivated additional experiments, such as analysing the effect on binding upon replacing E536 and R539 (Supporting Fig. S2C–E).

A key point for DrBphP is that Glu536 is central in the H+4 position, for the phosphatase activity argumentation as a key residue in DrBphP. The substitution to Ala is important, but also very heavy. I don't really understand why the authors don't explicitly analyze it in a bit more detail, even though they have addressed it themselves. I strongly suggest 2-3 more substitutions at the H+4 position to verify the argument, for example to Asp and to Gln or even to Thr or Asn (as in phosphatase activity in HisKA proteins). It would be very interesting and also important to show if the phosphatase activity can be somehow rescued. In general, the verification of the statements in the manuscript via mutational experiments for DrBphP is weak. There are only exactly three test mutations, but many important replacements that have not been made.

RESPONSE: In response, we generated additional residue exchanges at the H+4 position of DrBphP (E356D, E356N, E356Q, and E356T) and studied their impact on phosphatase activity (Fig. 3E). Whereas the E356A and E356Q variants retained phosphatase activity, it was lost in E356D, E356N and E356T, thus underlining the relevance of the H+4 site for the phosphatase reaction. Taken together, we thus consider our statement much better substantiated.

In contrast to DrRR, the response regulator AtRR1 could not be phosphorylated by acetyl-phosphate, as stated in the manuscript. Therefore, we have not been able to specifically assess phosphatase activity in Agp1 or its A532E variant, let alone the effect of other H+4 substitutions.

If the Agp1 concentration does not reach saturation (Fig.2), the KD value should probably not be reported. I also find it exceedingly difficult that the results of the follow-up experiments differ so much, which argues for not providing a KD determination, but simply saying that Agp1 binds weakly to At1RR1.

RESPONSE: We fully agree. As stated in our response to Reviewer #1, in the revised text we do not provide a concrete value for the Kd value of the Agp1-AtRR1 interaction but estimate the affinity to be on the order of hundreds of micromolar.

REVIEWERS' COMMENTS

Reviewer #4 (Remarks to the Author):

The authors have taken well into account many of the reviewers' concerns. The manuscript benefits from the reduction of MD data, which had nothing to do with the technical realization of these simulations, but rather with the very speculative nature of the data presented here. The idea of the manuscript and also the result is, as I had said earlier, very interesting and important to better understand differences in signaling between different bacterial phytochromes. And Agp1 and DrBphP in particular, as the best-studied model phytochromes, are ideally suited for this purpose. It is also a study that addresses a long-standing problem, namely what actually happens in the output modules of bacterial phytochromes. Except for one important point, I am very satisfied with the manuscript.

In the case of H+4, it seems clear that the charge of Glu just does not play a decisive role here, since the exchange against Gln seems to be the same, indeed it even shows a better phosphatase activity than the WT. Since (phospho-) gels always show different running characteristics, I think it would be very useful to show in Figure 3E the E536 mutants together with the WT in one gel.

By the way, it is wonderful that in the E536 substitution into Asp, Asn or Thr the smaller size of the amino acid plays a decisive role, since the phosphatase activity is completely inactive.

In contrast to the other variants (Asp, Asn, Thr), the variant to Ala still has a reduced phosphatase activity (which, however, does not quite correspond to that of WT or E536Q), which is somewhat surprising at first. Since charge does not seem to play a major role (see Gln), stabilization via H-bonds might be essential here, perhaps maintained by a water molecule in the case of the Ala mutant.

Line 557 is somewhat inaccurate and misleading in this context, in my opinion. The phosphatase activity is certainly somewhat lower than in WT and E536Q, but still quite evident (see above: Figure 3E should be done together with the WT).

Line 564 is really difficult. You have the same function with a Gln. Here you really have to keep in mind that the charge of the Glu does not play the essential role, but probably only the size and the possibility to form H-bonds (because of Gln).

The authors should discuss the above points in some more detail in the manuscript.

In this context, could phosphatase activity also be made pH dependent?

Minor point:

Figure 3B/line 247... What could the smeared signals in Figure 2b in Agp1 D529H variants actually mean, minor phosphorylation activity bands? One could mention that here in a side sentence.

REVIEWER COMMENTS

Reviewer #4 (Remarks to the Author):

The authors have taken well into account many of the reviewers' concerns. The manuscript benefits from the reduction of MD data, which had nothing to do with the technical realization of these simulations, but rather with the very speculative nature of the data presented here. The idea of the manuscript and also the result is, as I had said earlier, very interesting and important to better understand differences in signaling between different bacterial phytochromes. And Agp1 and DrBphP in particular, as the best-studied model phytochromes, are ideally suited for this purpose. It is also a study that addresses a long-standing problem, namely what actually happens in the output modules of bacterial phytochromes. Except for one important point, I am very satisfied with the manuscript.

RESPONSE: We thank the reviewer for the positive comments! We agree that the reduction of the MD data has improved the focus of the manuscript.

In the case of H+4, it seems clear that the charge of Glu just does not play a decisive role here, since the exchange against Gln seems to be the same, indeed it even shows a better phosphatase activity than the WT. Since (phospho-) gels always show different running characteristics, I think it would be very useful to show in Figure 3E the E536 mutants together with the WT in one gel.

RESPONSE: We agree that the inclusion of the WT sample in Figure 3e is important. We have now included the wild-type DrBphP in the Figure 3e.

By the way, it is wonderful that in the E536 substitution into Asp, Asn or Thr the smaller size of the amino acid plays a decisive role, since the phosphatase activity is completely inactive.

In contrast to the other variants (Asp, Asn, Thr), the variant to Ala still has a reduced phosphatase activity (which, however, does not quite correspond to that of WT or E536Q), which is somewhat surprising at first. Since charge does not seem to play a major role (see Gln), stabilization via H-bonds might be essential here, perhaps maintained by a water molecule in the case of the Ala mutant.

RESPONSE: We were also puzzled by the result that E536A still showed some phosphatase activity, and concur that this may be possible through water molecule(s) that adopt the position of the missing side chain. Since this is speculative, we refrain from mentioning this in the manuscript text.

Line 557 is somewhat inaccurate and misleading in this context, in my opinion. The phosphatase activity is certainly somewhat lower than in WT and E536Q, but still quite evident (see above: Figure 3E should be done together with the WT).

RESPONSE: We agree that mentioning that E536A mutation could remove the phosphatase activity is now misleading. We have therefore changed the sentence to “*Changing this residue to alanine diminishes ~~or removes~~ the phosphatase activity*”.

Line 564 is really difficult. You have the same function with a Gln. Here you really have to keep in mind that the charge of the Glu does not play the essential role, but probably only the size and the possibility to form H-bonds (because of Gln). The authors should discuss the above points in some more detail in the manuscript.

RESPONSE: We agree, and have now included the role of H-bond formation as a potential reason for activity: “*Notably, the glutamine at the H+4 position retains activity, indicating that its side-chain amide group likely retains similar H-bond interactions as the carboxylate group of the glutamate*”.

In this context, could phosphatase activity also be made pH dependent?

RESPONSE: We appreciate this incisive idea! Although we do not have currently any data on the effects of pH that could be added to the manuscript, this could be addressed in the future studies.

Minor point:

Figure 3B/line 247... What could the smeared signals in Figure 2b in Agp1 D529H variants actually mean, minor phosphorylation activity bands? One could mention that here in a side sentence.

RESPONSE: In Figure 2b, the smeared signal in Agp1 D529H variant corresponds to impurities in the sample preparation. This is evident by comparing the D529H-containing samples in Figure 2B and Supplementary Figure 4H. As the smears seem highly similar regardless of the assay conditions or illumination, we have excluded the possibility of phosphorylation activity.